# The interplay between randomness and structure during learning in RNNs

**Friedrich Schuessler**
Technion
schuessler@campus.technion.ac.il

**Francesca Mastrogiuseppe**
Gatsby Unit, UCL
f.mastrogiuseppe@ucl.ac.uk

**Alexis Dubreuil**
ENS Paris
alexis.dubreuil@gmail.com

**Srdjan Ostojic**
ENS Paris
srdjan.ostojic@ens.fr

**Omri Barak**
Technion
omri.barak@gmail.com

## Abstract

Recurrent neural networks (RNNs) trained on low-dimensional tasks have been widely used to model functional biological networks. However, the solutions found by learning and the effect of initial connectivity are not well understood. Here, we examine RNNs trained using gradient descent on different tasks inspired by the neuroscience literature. We find that the changes in recurrent connectivity can be described by low-rank matrices, despite the unconstrained nature of the learning algorithm. To identify the origin of the low-rank structure, we turn to an analytically tractable setting: training a linear RNN on a simplified task. We show how the low-dimensional task structure leads to low-rank changes to connectivity. This low-rank structure allows us to explain and quantify the phenomenon of accelerated learning in the presence of random initial connectivity. Altogether, our study opens a new perspective to understanding trained RNNs in terms of both the learning process and the resulting network structure.

## 1 Introduction

Recurrent neural networks (RNNs) have been used both as tools for machine learning, and as models for neuroscience. In the latter context, RNNs are typically initialized with random connectivity and trained on abstractions of tasks used in experimental settings [3, 20, 23, 32, 33, 35, 37, 40]. The obtained networks are then compared to both behavioral and neural experimental results, with the added advantage that the RNNs are more amenable to analysis than their biological counterparts [34]. Despite this advantage, the understanding of how RNNs implement neuroscience tasks is still limited. Open questions concern especially the relationship between the final connectivity and the task, and its formation through training.

Here, we examine the relation between the initial connectivity of the RNN, the task at hand, and the changes to connectivity through training. We use unconstrained gradient descent that can potentially alter the connectivity completely. However, evaluating nonlinear RNNs trained on several neuroscience-inspired tasks, we observe that the connectivity changes are small compared to the initial connectivity. We thus split the connectivity matrix $W$ at the end of training into the initial part $W_0$ and the changes $\Delta W$, writing

$$W = W_0 + \Delta W .\tag{1}$$

For all tasks we consider, we find that the training-induced connectivity structure $\Delta W$ is of low rank, despite the unconstrained nature of training used. This finding directly connects gradient-based learning with a number of existing neuroscience frameworks based on low-rank aspects of connectivity

[4, 9, 12, 14, 18, 21, 24, 33, 36]. Despite the low-rank nature of the *changes* to connectivity $\Delta W$, the initial, full-rank, random connectivity $W_0$ plays an important role in learning. Consistent with previous work [28, 33], we find that the initial connectivity accelerates learning. Moreover we show that the final, trained network relies on correlations between $\Delta W$ and $W_0$.

In the second part of our work, we analyze the mechanism behind these observations in a simplified and analytically tractable setting: nonlinear dynamics of learning in a linear RNN trained on a simple input-output mapping task. We show how the low-dimensional task structure leads to low-rank connectivity changes; importantly, the amplitude and geometry of these low-rank changes depend on the random initial connectivity. Our work reveals how this dependence accelerates learning and quantifies the degree of acceleration as a function of initial connectivity strength.

Finally, we show that our results extend to real-world settings of an LSTM network trained on a natural language processing task, suggesting practical applications of our results.

## 2 Training RNNs on low-dimensional tasks

**Tasks**   We trained RNNs on three tasks inspired by the neuroscience literature. All tasks are characterized by a small number of input and output channels. The first task is a working memory task, in which the network receives pulses from two different input channels and needs to remember the sign of the last pulse in each channel independently [34]. The second task is a context-dependent decision task: The network receives two noisy signals, as well as one of two context inputs which indicates the relevant signal. After the input presentation, it needs to output whether the average of the relevant signal was positive or negative [20]. The third task is a delayed-discrimination task [25] in which the network receives two positive pulses separated by a delay. After yet another delay, it needs to output which of the two pulses had the larger amplitude. Based on their origin, we refer to the three tasks as "flip-flop" [34], "Mante" [20], and "Romo" [25] task, respectively. For each task, we plotted a single trial for a successfully trained network in Fig. 1(**a-c**). Detailed parameters can be found in the supplementary.

**RNN model**   Each RNN model consists of $N$ neurons whose state vector evolves according to

$$\dot{\mathbf{x}}(t) = -\mathbf{x}(t) + W\phi(\mathbf{x}(t)) + \sqrt{N}\sum_{i=1}^{N_{\text{in}}} \mathbf{m}_i u_i(t). \qquad (2)$$

The recurrent input is given by the firing rate vector $\phi(\mathbf{x})$ multiplied by the weight matrix $W$. We use the element-wise nonlinearity $\phi = \tanh$. The network receives time-dependent inputs $u_i(t)$ through input vectors $\mathbf{m}_i$. The output is the projection of the firing rate onto readout vectors $\mathbf{w}_i$, namely

$$z_i(t) = \frac{\mathbf{w}_i^T\phi(\mathbf{x}(t))}{\sqrt{N}} \quad \text{for } i \text{ in } \{1,\ldots,N_{\text{out}}\}. \qquad (3)$$

We formulate target values $\hat{z}_i(t)$ during specific segments of the trial [see dark lines for output panels in Fig. 1(**a-c**)]. The task determines the numbers $N_{\text{in}}$ and $N_{\text{out}}$ of input and output vectors. For example, the Mante task requires four input vectors (for both signals and contexts) and a single output vector. We are interested in the behavior of large networks, $N >> 1$, while the dimension of the tasks is small, $N_{\text{in}}, N_{\text{out}} \sim \mathcal{O}(1)$. For the simulation, we chose $N$ to be large enough so that learning dynamics become invariant under changes in $N$ (see supplementary Fig. S1).

**Training and initialization**   For training the RNNs, we formulated a quadratic cost in $z_i(t)$ and applied the gradient descent method "Adam" [15] to the internal connectivity $W$ as well as to the input and output vectors $\mathbf{m}_i$, $\mathbf{w}_i$. Restricting the updates to $W$ or training with SGD impaired the convergence times but yielded similar results (not shown). The initial input and output vectors were drawn independently from $\mathcal{N}(0, 1/N)$. We initialized the internal weights as a random matrix $W_0$ with independent elements drawn from $\mathcal{N}(0, g^2/N)$. The parameter $g$ thus scales the strength of the initial connectivity.

**Learning dynamics in the absence of initial connectivity**   To understand what kind of connectivity arises during learning, we first looked at the simplest case without initial connectivity, $g = 0$. The loss curves indicate convergence for all three tasks [see darker lines in Fig. 1(**d-f**)]. We analyzed the

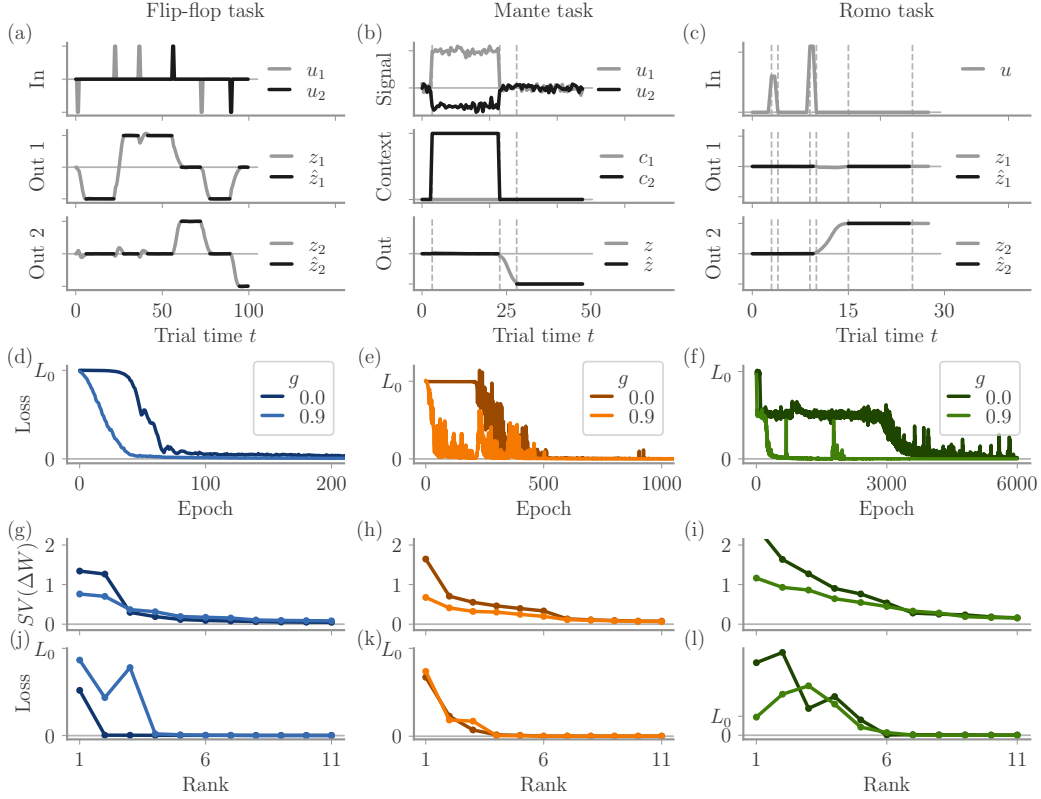

Fig. 1: Learning dynamics in three different neuroscience tasks. **(a-c)** Task summary: inputs $u_i$, outputs $z_i$, and targets $\hat{z}_i$ for each task. Dashed lines indicate task phases. **(d-f)** Loss throughout training process for different initial connectivity strengths $g$. $L_0$ is the loss at the beginning of training for $g = 0$ ($L_0$ is different for different tasks). Note the different epoch numbers plotted. **(g-i)** First 11 singular values of final connectivity changes $\Delta W$. **(j-l)** Loss for truncated networks, where $\Delta W$ is replaced with the rank-$R$ approximation $\Delta W^{(R)}$. Parameters: $N = 256$, learning rate $\eta = 0.05/N$.

connectivity at the end of training by computing its singular values (SVs). For the flip-flop task, we found that the first two SVs were much larger than the remaining ones [Fig. 1**(g)**]. To see whether the network utilizes this approximate rank-two structure, we replaced the changes $\Delta W$ with the singular value decomposition truncated at rank $R$,

$$\Delta W^{(R)} = \sum_{r=1}^{R} s_r \mathbf{u}_r \mathbf{v}_r^T \ . \tag{4}$$

Note that we keep the initial connectivity $W_0$. The loss after truncation indeed drops to zero at rank 2 [Fig. 1**(j)**]. A similar situation is observed for the Mante and Romo tasks, see Fig. 1**(h, k)** and **(i, l)**, respectively. Although for these tasks the SVs drop more slowly, the first six SVs are discernibly larger than the remaining tail; the truncation loss drops to zero at rank 4 and 6, respectively. In sum, we observe that for $g = 0$, training via gradient descent yields an effective low-rank solution for all three tasks.

**Effects of initial connectivity on learning dynamics and connectivity** The loss-curves in Fig. 1**(d-f)** indicate a strong influence of the initial connectivity strength $g$ on the training dynamics (lighter colors for $g = 0.9$). We observe that learning becomes faster and smoother with initial connectivity. In Fig. 2**(a)**, we quantify the acceleration of learning with the number of epochs needed to reach 5% of the initial loss. We observe that convergence time smoothly decreases as a function of connectivity strength g; for very large g, networks finally transition to chaotic activity [31], and convergence time increases again.

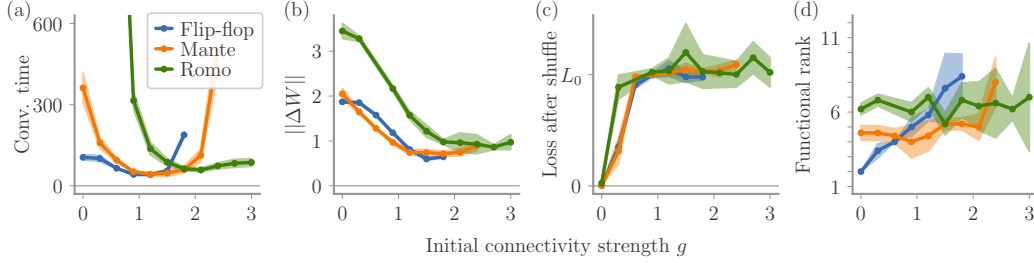

Fig. 2: Dependence of learning dynamics on initial connectivity strength $g$ in the three tasks. Lines and shades indicate mean and standard deviation of five independent simulations for each $g$, respectively. **(a)** Number of epochs at which the loss falls below 5% of $L_0$. **(b)** Frobenius norm of $\Delta W$ at the end of training. **(c)** Loss for shuffled initial connectivity, so that the full network connectivity is given by $W_0^{\text{shuffle}} + \Delta W$. **(d)** Rank $R$ at which the loss of the network with rank-truncated connectivity $\Delta W^{(R)}$ drops below 5% of the initial loss $L_0$.

After observing the drastic decrease in learning time, we wondered how initial connectivity affects the resulting connectivity changes. The first observation is that, for increasing $g$, the final connectivity $W = W_0 + \Delta W$ is dominated by $W_0$, since $||W_0|| = \sqrt{N}g$. In fact, the norm of $\Delta W$ not only remains unchanged for increasing $N$ (see supplementary), but further decreases with increasing $g$, see Fig. 2**(b)**. If a smaller $\Delta W$ solves the task for larger initial connectivity, it is reasonable to assume that $W_0$ amplifies the effect of $\Delta W$. To test this idea, we shuffled the elements of $W_0$, destroying any correlation between $W_0$ and $\Delta W$, while maintaining its statistics. The loss after replacing the connectivity with $W_0^{\text{shuffle}} + \Delta W$ is shown in Figure 2**(c)**. For all tasks, shuffling strongly degraded performance except for cases with very weak initial connectivity.

**Low-rank changes in connectivity** Despite the effects of the initial connectivity on convergence time and the norm of $\Delta W$, the low-rank nature of $\Delta W$ remains similar to the case with $g = 0$. In Fig. 1**(g-h)**, the SVs of $\Delta W$ are plotted in lighter colors. We see that the pattern and overall amplitude is very similar to the darker lines for $g = 0$: only a small number of SVs dominates over a tail. To assess the functional rank, we replaced $\Delta W$ in our RNN with the rank-$R$ truncation, Eq. (4), while keeping the initial connectivity $W_0$ identical. The resulting loss, Fig. 1**(j-l)**, indicates that the effective connectivity change is indeed low-rank: for all three tasks, it drops to a value close to zero before rank 10. We quantified this observation by computing the "functional rank", the rank at which the loss decreases below 5% of the initial value [see Fig. 2**(d)**]. This functional rank is between 2 and 10 for all three tasks (averaged over independent simulations). It increases with $g$ for the flip-flop task, while it remains less affected for the other two tasks.

## 3 Analytical results for linear system

The observation of effective low-rank changes in connectivity and accelerated learning for random initial connectivity were general across the three different tasks considered. To understand the underlying mechanisms, we turn to a much simpler task and a linear RNN model. This setting allows us to analytically describe the learning dynamics, understand the origin of the low-rank connectivity changes, and quantify how correlations between $W_0$ and $\Delta W$ accelerate learning. Our approach is similar to that of Saxe et al. [27], who analyzed gradient descent dynamics in linear feed-forward networks. Both for the feed-forward and the recurrent model, the learning dynamics are nonlinear despite the linearity of the networks. Nevertheless, we will see that the recurrent nature of our models results in very different dynamics compared to the linear feed-forward model. Below we will present our main results for the simplified model; the details of all our analytical derivations can be found in the supplementary.

**Simplified setting** Our simple task is an input-output transformation: Given a constant input $u(t) = 1$, the output $z(t)$ has to reach a target value $\hat{z}$ at time $T$. The corresponding loss is $L = (\hat{z} - z(T))^2/2$. An example with two different target values $\hat{z} = 0.5,\ 2.0$ is plotted in Fig. 3**(a)**. The linear RNN model is obtained by replacing the nonlinearity in Eq. (2) with the identity, $\phi(\mathbf{x}) = \mathbf{x}$,

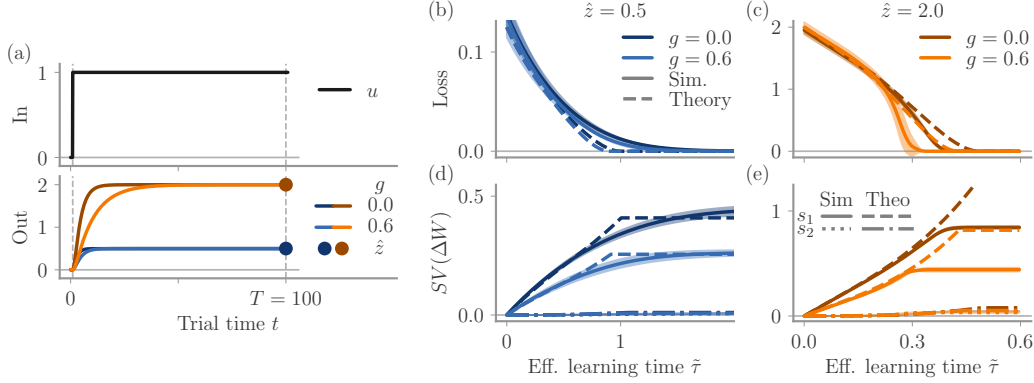

Fig. 3: Learning a simple input-output transformation in a linear network. **(a)** Task summary. Output for trained networks with two different initial connectivity strengths $g = 0.0$, $0.6$ and target amplitudes $\hat{z} = 0.5$, $2.0$. Input starts at $t = 1$, loss is evaluated at $T = 100$. **(b,c)** Loss over training for target values $\hat{z} = 0.5$ and $\hat{z} = 2.0$. Full lines indicate simulation results, dashed lines our theoretical prediction. **(d,e)** First two SVs of $\Delta W$ at the end of training (full, dotted lines) and theoretical predictions (dashed, dashed-dotted). In panels **(b-e)**, the simulation results are averaged over five independent instances. Shades, if visible, indicate the standard deviation. Note that the x-axes in **(b-e)** show the rescaled, effective learning time $\tilde{\tau} = \beta^2 \tau$, with $\beta = 1/(1 - g^2)$. Simulation parameters: $N = 1024$, training for 200 epochs with learning rate $\eta$ adapted (see supplementary).

and keeping only a single input and output. All weights are initialized as before. We keep the initial connectivity strength $g < 1$ so that the linear network remains stable. To further simplify, we constrain weight changes to the recurrent weights $W$ only, and apply plain gradient descent. To compare between different simulations, we define the learning time $\tau = \eta \cdot$ epochs.

Evaluating the trained networks reveals similar phenomena as observed for the nonlinear, more complex tasks. Figure 3(**b-e**) shows the loss and SVs of $\Delta W$ over learning time for two values of $g$. We observe that learning induces low-rank connectivity changes – in fact, a single SV dominates. Because of the small magnitude of the second SV, truncating $\Delta W$ at rank 1 does not lead to increased loss (not shown), so that the functional rank as defined in the previous section is 1. Comparing between $g = 0$ and $g = 0.6$, we further see that learning is accelerated by the initial connectivity, and that the magnitude of the first SV decreases with increasing $g$. These observations will be quantified with our analytical results.

**Gradient descent dynamics** For our analytical treatment, we only consider the limit of long trials, with the output $z = \lim_{T \to \infty} z(T)$ at the end of a trial. In this limit, the network converges to its fixed point $\mathbf{x}^* = \sqrt{N} \left( I - W \right)^{-1} \mathbf{m}$ with identity matrix $I$, and the readout is

$$z = \frac{\mathbf{w}^T \mathbf{x}^*}{\sqrt{N}} = \mathbf{w}^T (I - W)^{-1} \mathbf{m}. \tag{5}$$

The input and output vectors, $\mathbf{m}$ and $\mathbf{w}$, remain fixed during training, and only $W$ is changed. We can explicitly compute the changes induced by the gradient of the loss:

$$\frac{dW(\tau)}{d\tau} = -\frac{dL}{dW} = [\hat{z} - z(\tau)] \left[ I - W^T(\tau) \right]^{-1} \mathbf{w}\mathbf{m}^T \left[ I - W^T(\tau) \right]^{-1}, \tag{6}$$

with initial connectivity $W(0) = W_0$. We made a continuous-time approximation of the weight updates ("gradient flow"), valid to small learning rates $\eta$. Note that the readout $z$ at the fixed point depends on the learning time $\tau$ through $W(\tau)$.

Note that, unlike the feed-forward case [26], the inverse of $W$ appears in Eq. (6), opening the possibility of divergence during learning. It also precludes a closed-form solution to the dynamics. However, we can obtain analytical insight by expanding the learning dynamics in learning time around the initial connectivity [5]. We write

$$W(\tau) = \sum_{k=0}^{\infty} W_k \frac{\tau^k}{k!}. \tag{7}$$

The changes in connectivity are obtained by subtracting $W_0$, which yields $\Delta W(\tau) = W_1 \tau + W_2 \tau^2/2 + \dots$. We analytically computed the coefficients $W_k$ by evaluating $\mathrm{d}^k W/\mathrm{d}\tau^k$ at $\tau = 0$. A comparison of the expansion up to third order with the numerical results from gradient descent learning indicates close agreement during most of the learning [see Fig. 3(**b-e**) full vs. dashed lines].

**Learning dynamics in absence of initial connectivity**    It is instructive to first consider the case of no initial connectivity, $g = 0$. The readout at the beginning of training is then $z_0 = \mathbf{w}^T \mathbf{m}$. Due to the independence of $\mathbf{m}$ and $\mathbf{w}$, the expected value of $z_0$ vanishes. Moreover, the standard deviation scales as $1/\sqrt{N}$ with the network size. In this work, we are interested in the learning dynamics for large networks; all our analytical results are valid in the limit $N \to \infty$. We therefore write $z_0 = 0$. Similar reasoning goes for all scalar quantities of interest: they are of order $\mathcal{O}(1)$, with deviations $\mathcal{O}(1/\sqrt{N})$. With this self-averaging quality, we omit stating the limit as well as the expectation symbol and use the equality sign instead.

Inserting $W_0$ and $z_0$ – both zero – into the gradient descent, Eq. (6), yields the first order coefficient

$$W_1 = \hat{z}\mathbf{w}\mathbf{m}^T. \tag{8}$$

Hence, the weight changes at linear order in $\tau$ are described by a rank-one matrix, and the readout is $z(\tau) = \tau \hat{z} + \mathcal{O}(\tau^2)$. The gradient descent for $g = 0$ would therefore converge at $\tau_1^* = 1$, if it only depended on the first-order term. The numerical results already show deviations in the form of faster or slower convergence, depending on the target $\hat{z}$ [see dark lines in Fig. 3(**b,c**) and note that $\tilde{\tau} = \tau$ for $g = 0$]. This indicates the importance of higher order terms.

We observe that the gradient in Eq. (6) contains the transpose $W^T$. At higher orders, this term introduces other outer-product combinations of $\mathbf{m}$ and $\mathbf{w}$. In fact, for $g = 0$, these are the only vectors present in the gradient, so that the connectivity can always be written as

$$\Delta W(\tau) = \begin{bmatrix} \mathbf{w} & \mathbf{m} \end{bmatrix} \begin{bmatrix} A_{11} & A_{12} \\ A_{21} & A_{22} \end{bmatrix} \begin{bmatrix} \mathbf{w}^T \\ \mathbf{m}^T \end{bmatrix}. \tag{9}$$

This form implies that $\Delta W$ will be at most a rank-two matrix. An analysis of the SVs [Eq. (14) below for general $g$] reveals that the second SV remains very small, as visible in Fig. 3(**d,e**).

The entries of the $2 \times 2$ matrix $A(\tau)$ up to order $\mathcal{O}(\tau^3)$ are (see supplementary)

$$A_{11} = \frac{\hat{z}^2}{2}\left(\tau^2 - \tau^3\right), \qquad A_{12} = \hat{z}\left(\tau - \frac{\tau^2}{2} + \frac{\tau^3}{6}(1 + 2\hat{z}^2)\right), \qquad A_{21} = \frac{\hat{z}^3 \tau^3}{3}, \tag{10}$$

and $A_{22} = A_{11}$. The first surprising observation is that the target value $\hat{z}$ enters nonlinearly into the expressions above. This is the origin of the qualitative difference between learning curves for different values of the target output in Fig. 3(**b,c**).

We further observe that the connectivity changes develop a nonzero eigenvalue only at $\mathcal{O}(\tau^2)$. This is because the off-diagonal terms, which grow linearly with $\tau$ contribute a zero eigenvalue because $\mathbf{m}^T \mathbf{w} = 0$. At second order the diagonal entries of $A$ – and, with it, the eigenvalues – change. Changes in connectivity eigenvalues imply changes in time scales of network dynamics, which may be necessary for some tasks (for example, those involving memory), but can also lead to problems of exploding gradients (see supplementary).

**Effects of initial connectivity**    In the presence of initial connectivity, we can still apply the expansion introduced above. Due to the independence of $W_0$, $\mathbf{m}$, and $\mathbf{w}$, the initial readout $z_0$ remains zero. The gradient descent, Eq. (6), then directly yields the first-order connectivity coefficient

$$W_1 = \hat{z}\,B^T\mathbf{w}\,\mathbf{m}^T B^T, \qquad \text{with} \qquad B = (I - W_0)^{-1}. \tag{11}$$

Thus, $W_1$ is still a rank-one matrix despite the full-rank initial connectivity. However, the connectivity changes now include the initial connectivity $W_0$ via the matrix $B$. As a consequence, the norm of the first-order coefficient, $||W_1|| = \hat{z}\beta$ (see supplementary), increases with $g$ by the factor

$$\beta = \mathbf{w}^T B B^T \mathbf{w} = \mathbf{m}^T B^T B \mathbf{m} = \frac{1}{1 - g^2}. \tag{12}$$

The readout is also affected by the initial connectivity. We compute (see supplementary)

$$z(\tau) = \tau \hat{z} \beta^2 + \mathcal{O}(\tau^2). \tag{13}$$

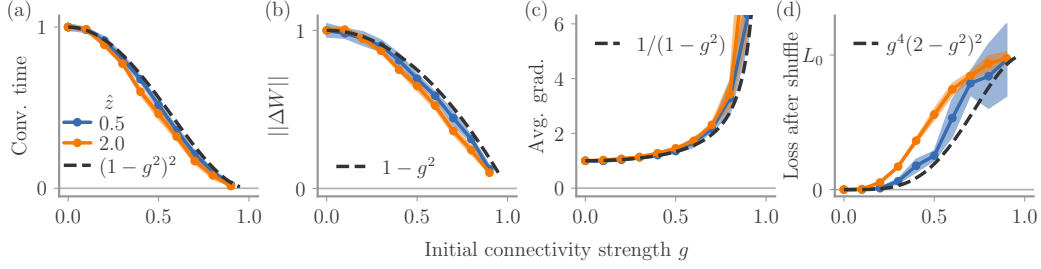

Fig. 4: Dependence of learning dynamics on initial connectivity strength $g$ in the simplified task. **(a)** Learning time $\tau^*$ until loss reached 5% of its initial value. **(b)** Norm of final weight changes $\Delta W$. **(c)** Norm of gradient $dW/d\tau$, averaged over the interval $\tau \in [0, \tau^*]$. The quantities in panels **(a-c)** are normalized by their value at $g = 0$. **(d)** Loss after shuffling the initial connectivity $W_0$, normalized by initial loss. In all panels, thick full lines indicate the average over five independent simulations, shades the standard deviation, and dashed lines the first-order theoretical prediction.

Learning converges when $z(\tau)$ reaches the target value $\hat{z}$. The first-order prediction of the convergence time is therefore $\tau_1^* = 1/\beta^2$, and the initial connectivity accelerates learning by the factor $1/\beta^2 = (1-g^2)^2$. We can decompose this acceleration into two factors: The growth rate is increased by $\beta$, and the norm of the final connectivity changes decreased by $1/\beta$. For the first contribution, we note that the first-order coefficient $W_1$ is, by definition, the constant part of the gradient, and hence the rate at which connectivity changes. For the second contribution, we compute the norm of $\Delta W(\tau)$ at the predicted convergence time $\tau_1^*$ (see supplementary).

In Fig. 4**(a-c)**, we compare our first-order predictions with numerical simulations. In panels **(a,b)**, we plot the convergence time $\tau^*$ and the norm of $\Delta W$ at the end of training. As for the more complex, nonlinear tasks [see Fig. 2**(a,b)**], we defined the numerical $\tau^*$ as the point in time where the loss drops to 5% of the initial value. For the gradient, panel **(c)**, we averaged the norm $||dW/d\tau||$ over the interval $[0, \tau^*]$. To compare the collapsed curves with the predicted scalings, we normalized the curves for the different target values $\hat{z}$ by their value at $g = 0$ for all three quantities. We observe good agreement between the numerical results and the theory, even though we only used the first-order predictions, and $\tau^*$ often shows notable differences between theory and simulation [for example in Fig. 3**(b,c)**].

Finally, we assess the role of correlations between $\Delta W$ and $W_0$ by shuffling $W_0$. After shuffling, the readout loses the amplification by $\beta^2$ and is hence $z^{\text{shuff}} = \tau_1^* \hat{z}$. The corresponding loss is $L_1^{\text{shuff}} = L_0 \, g^4(2-g^2)^2$, with initial loss $L_0 = \hat{z}^2/2$. A comparison of this first-order prediction with numerical results shows qualitative agreement with notable quantitative differences especially for the larger target amplitude, see Fig. 4**(d)**. A comparison with the nonlinear case, Fig. 2**(c)** shows that our simple model captures the phenomenon qualitatively.

**Higher-order terms**  Does the initial connectivity lead to higher-rank changes in connectivity? For $g > 0$, the explicit rank-two expression for the weight changes, Eq. (9), does not hold anymore: The input and output vectors accumulate multiples of $B$ and $B^T$ (such as $B^T\mathbf{w}$ and $BB^T\mathbf{w}$) which increase the number of possible outer products – and hence potentially the rank. However, computing the first two SVs, $s_1$ and $s_2$, up to order $\mathcal{O}(\tau^3)$ (see supplementary) shows that $\Delta W$ remains approximately rank one:

$$s_1 = \frac{\hat{z}}{\beta}\left[\tilde{\tau} - \frac{\tilde{\tau}^2}{2} + \left(1 + \frac{7}{2}\hat{z}^2\beta\right)\frac{\tilde{\tau}^3}{6}\right], \qquad s_2 = \hat{z}^3\frac{\tilde{\tau}^3}{12}. \tag{14}$$

where $\tilde{\tau} = \beta^2\tau$ is the effective learning time. We observe that $s_1$ grows linearly, but $s_2$ only at third order of $\tau$. Different parts of connectivity therefore grow on top of each other, giving rise to a temporal hierarchy in the learning dynamics. Numerical simulations show good agreement with this prediction (see supplementary).

We further state the resulting readout up to $\mathcal{O}(\tau^3)$:

$$z(\tau) = \hat{z}\left[\tilde{\tau} - \frac{\tilde{\tau}^2}{2} + (1 + 8\hat{z}^2\beta)\frac{\tilde{\tau}^3}{6}\right]. \tag{15}$$

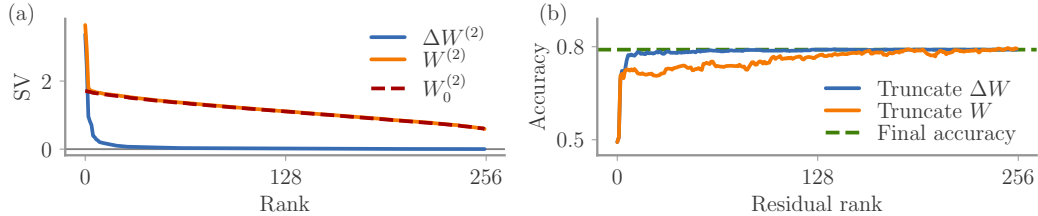

Fig. 5: Low-rank changes for a two-layer LSTM model trained on a sentiment analysis task. **(a)** Singular values (SVs) of the recurrent weights in the second layer (256 neurons). The initial, random $W_0$ is full rank, and the final $W$ visibly differs from it only for the first SVs. The changes, $\Delta W$, are approximately low-rank. **(b)** Validation accuracy after truncating the lower singular values of connectivity. We either truncated $W$ directly, or applied truncation only to $\Delta W$ while keeping $W_0$. We truncated the recurrent weights of both layers and the input weights of layer 2.

The appearance of $\beta$ in the third-order contributions in Eqs. (14) and (15) shows that the learning with different values of $g$ does not entirely collapse onto one curve after rescaling the time by $\beta^2$. Instead, there is an additional acceleration, which increases with increasing target amplitude $\hat{z}$. This effect can be appreciated in Fig. 3**(b,c)**, where for larger $\hat{z}$ the loss curve becomes concave. Note that our approximation up to $\mathcal{O}(\tau^3)$ predicts this trend, despite quantitative disagreement. As we saw in Fig. 4, the scaling of the convergence time $\tau^*$ with $g$ is not strongly affected by the higher order terms.

## 4 Beyond neuroscience tasks

We asked whether our observation that connectivity changes are low-rank despite full-rank initial connectivity would extend to more complex network architectures and tasks, specifically those not restricted to a small input or output dimension. We therefore trained a two-layer LSTM network on a natural language processing task, sentiment analysis of movie reviews [30] (details in supplementary).

The SVs at the end of training showed the pattern that we predicted: learning only leads to small changes in the connectivity so that the final connectivity $W$ is dominated by the initial connectivity and has full rank. The changes $\Delta W$ only have a small number of large SVs. For the recurrent weights of layer 2, the SVs are plotted in Fig. 5**(a)**; other weights behave similarly (see supplementary).

Like before, we evaluated the accuracy of networks after truncation at a given rank, see Fig. 5**(b)**. We truncated the recurrent weights of both layers as well as input weights to layer 2. If we keep the random parts and truncate the changes as in Eq. (4) a rank-10 approximation already yields the final training accuracy. In contrast, if we truncate the entire weight matrices, as previously suggested [38], it takes more that half of the network rank (256 neurons per layer) to get close to the final accuracy.

## 5 Discussion

**Summary of results** Our key finding is that the connectivity changes $\Delta W$ induced by unconstrained training on low-dimensional tasks are of low rank. With our simplified analytical model, we demonstrated why: The connectivity changes are spanned by a small number of existing directions, determined by the input and output vectors. Without initial connectivity, the maximum rank that linear networks can obtain through learning is in fact bounded by this number. The initial connectivity $W_0$ enlarges the pool of available directions. The fact that learning arrives at a low-rank solution even in presence of initial connectivity is then a result of the temporal structure of learning: Initially, only a small number of available directions grow, inducing a low-rank structure. For our simplified task, the first of these structures already reduces the loss, and learning converges before other structures emerge; the final connectivity changes are hence rank-one. For other tasks, the available input and output directions alone may not be sufficient, so that initial connectivity becomes necessary for successful learning (see supplementary). Note that our theoretical analysis is limited to linear networks; however, nonlinearity may also contribute to generate novel learning directions.

Our numerical simulations further showed that initial connectivity significantly accelerated learning. Our analytical results revealed the underlying mechanism: The input and output vectors spanning the gradient are multiplied by powers of $W_0$, which strongly correlates $\Delta W$ to $W_0$. This correlation amplifies the effect of $\Delta W$, and removing the correlation by shuffling $W_0$ indeed degrades performance. This is in line with a recent study demonstrating such amplification through correlation between a random matrix and a low-rank perturbation in a model without learning [29].

Finally, we showed that the general observation of low-rank weight changes indeed holds even in a much more complex setting: a sentiment analysis task and a two-layer LSTM network. This implies a large potential for network compression [38]: one may truncate the changes in connectivity at a very low rank and recover the specific random initial connectivity using the seed of its random number generator.

**Task dimension and rank** Low-rank connectivity structures have previously been studied and applied. On the one hand, a number of RNN frameworks explicitly rely on low-rank feedback for training [4, 9, 14, 18, 33]. On the other hand, low-rank networks are amenable to analysis, because the network activity is low-dimensional and evolves in directions determined by the vectors spanning the connectivity [12, 21, 24, 29, 36]. Our surprising observation that unconstrained gradient descent also leads to low-rank connectivity opens new possibilities for studying general gradient-based learning with the tools developed by previous works.

We observed that the functional rank of the training-induced connectivity changes is strongly task dependent. A better understanding of the relation between task and connectivity calls for a concept of a task dimension, ideally based on the underlying abstract computations and independent of the specific implementation [10, 17, 19, 40]. Such a concept would allow to compare the solutions obtained by different algorithms and define a necessary minimal rank for a given task [8].

**Learning as a dynamical process and relation to feed-forward networks** Our approach stresses a dynamical perspective on learning, in which the solutions are not determined by the task alone, but also by the initial connectivity and the temporal evolution of weight changes. In particular, our expansion in learning time shows that some components in the connectivity only grow after others are present, which induces a temporal hierarchy. This affects the solutions the network arrives at. The temporal structure may also induce pitfalls for learning, for example divergent gradients when the networks undergo a phase transition [22] (see supplementary). A better understanding of the learning dynamics could be used to circumvent such problems, for example by introducing adapted learning curricula [6].

Learning in feed-forward networks has previously been analyzed from a similar perspective. It was found that the statistical structure of the training data induces a temporal hierarchy with long plateaus between step-like transitions in the learning curve [1, 11, 16, 26, 27, 41]. The hierarchy in our work originates in the dynamics of the RNN rather than the structure of the training data. For example, the plateaus seen in Fig. 1(**d-f**) can be related to phase transitions in the network dynamics, such as the emergence of new fixed points. Combining such internal learning dynamics with structured training data would be an interesting future direction.

Finally, recent work on feed-forward networks identified two different learning regimes: a kernel regime vs. a rich, feature-learning regime [2, 7, 13, 39]. In the prior, the change in weights vanishes as the network width increases, and the network function can be linearized around the weights at initialization. In our work, too, the weight changes $\Delta W$ become infinitely small in the limit of wide networks. However, even such vanishing $\Delta W$ may significantly change the dynamics of the neural network by inducing large outlier eigenvalues [29]. For example, the readout for our linear network, Eq. (5), diverges for a eigenvalue of $W$ approaching 1. In such a case, the network function cannot be approximated by linearization around the initial weights. Understanding the relation between learning regimes in feed-forward and recurrent networks constitutes an interesting field for future studies.

## Broader Impact

This work is a theoretical study on the dynamics of learning in RNNs. We show which kind of connectivity changes are induced by gradient descent. We expect that our insights will help to understand learning in RNNs, which benefits the research community as a whole and may ultimately lead to the development of improved learning algorithms or schemes. As a possible application, we show that one can use our results to efficiently compress a multi-layer RNN trained on a natural language processing task. In this work, there are no new algorithms, tasks, or data sets introduced. Therefore, the questions regarding any disadvantages, failures of the system, or biases do not apply.

## Acknowledgments and Disclosure of Funding

This work was supported in part by the Israeli Science Foundation (grant number 346/16, OB). The project was further supported by the ANR project MORSE (ANR-16-CE37-0016), the program "Ecoles Universitaires de Recherche" launched by the French Government and implemented by the ANR, with the reference ANR-17-EURE-0017. F.S. acknowledges the Max Planck Society for a Minerva Fellowship. There are no competing interests.

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
