[Supplementary Material]

# Supplementary information

## 1 Simulation parameters

All simulations were based on pytorch [5]. For the nonlinear neuroscience tasks, we applied the gradient descent method "Adam" [4] to the recurrent weights $W$ as well as to the input and output vectors $\mathbf{m}_i$, $\mathbf{w}_i$. We checked that our results did not depend qualitatively on the choice of the "Adam" algorithm over plain gradient descent; however, training converged more easily for this choice of algorithm. We also checked that restricting training to $W$ only (as for the simple model) did not alter our results qualitatively (although, with this restriction, training on the Romo task for small values of $g$ did not converge). Code for reproducing our results can be found on https://github.com/frschu/neurips_2020_interplay_randomness_structure/.

The network size for the results in Figures 1 and 2 was $N = 256$, and the learning rate $\eta = 0.05/N$. We trained the networks for a maximum number of 1000, 2000, and 6000 epochs for the flip-flop, Mante, and Romo task, respectively. Each epoch consisted of a batch of 32 independent task trials. For evaluation of the loss after rank-truncation or shuffling $W_0$, we used a single batch of 512 independent task trials. Note that for "Adam", the learning rate is scaled with $N$ to obtain approximate invariance of the loss curve for different network sizes $N$. Further note that Fig 1 does not always show the loss over all learning epochs (so that the differences in the initial phase are more clearly visible).

For the simpler, linear model, we applied plain gradient descent and only adapted $W$. We trained all models for 200 epochs, and the learning rate was adapted in order to obtain smooth convergence within these 200 epochs. We set $\eta = \eta_0(1 - g^2)^2$, with $\eta_0 = 0.015, 0.003$ for $\hat{z} = 0.5, 2.0$, respectively. We checked that our numerical results do not depend on this choice, as long as a sufficiently small learning rate and large enough number of epochs is chosen.

The network dynamics are described by the continuous dynamics

$$\dot{\mathbf{x}}(t) = -\mathbf{x}(t) + W\phi(\mathbf{x}(t)) + \sqrt{N}\sum_{i=1}^{N_{\text{in}}} \mathbf{m}_i u_i(t),\tag{1}$$

with initial condition $\mathbf{x}(0) = \mathbf{0}$. For the simulation, we discretized these using the Euler-forward scheme:

$$\mathbf{x}_{k+1} = (1 - \Delta t)\mathbf{x}_k + \Delta t\left[W\phi(\mathbf{x}_k) + \sqrt{N}\sum_{i=1}^{N_{\text{in}}} \mathbf{m}_i u_{i,k}\right],\tag{2}$$

with a discrete time step $\Delta t = 0.5$ and $\mathbf{x}(t = k\Delta t) = \mathbf{x}_k$. We checked that our results did not change qualitatively for choosing a smaller $\Delta t$ or fully discrete dynamics ($\Delta t = 1$).

For the gradient-based updates, we defined the quadratic loss

$$l(t) = \frac{1}{N_{\text{out}}}\sum_{i=1}^{N_{\text{out}}} \frac{1}{2}\left[z_i(t) - \hat{z}_i(t)\right]^2,\tag{3}$$

with readout $z_i(t)$, target $\hat{z}_i(t)$, and number of outputs $N_{\text{out}}$. Depending on the task, the loss was defined only during specific times of the task (during decision or fixation periods, see task descriptions). Accordingly, for each task we defined a boolean mask $M_k$, indicating the points $k$ on the discrete time grid were the loss was active. The full loss was the average over this mask:

$$L = \frac{1}{N_M}\sum_{k=1}^{k_{\text{max}}} M_k\, l(k\Delta t),\tag{4}$$

with $N_M = \sum_{k=0}^{k_{\text{max}}} M_k$, $k_{\text{max}} = T/\Delta t$ and trial time $T$.

Table S1: Task parameters

| Parameter | Symbol | Flip-flop | Mante | Romo | Simple task |
|---|---|---|---|---|---|
| # inputs | $N_{\text{in}}$ | 2 | 4 | 1 | 1 |
| # outputs | $N_{\text{out}}$ | 2 | 1 | 2 | 1 |
| Trial duration | $T$ | 50 | 48 | 30 | 101 |
| Fixation duration | $t_{\text{fix}}$ | 1 | 3 | 3 | 1 |
| Stimulus duration | $t_{\text{stim}}$ | 1 | 20 | 1 | - |
| Decision delay | $t_{\text{delay}}$ | 5 | 5 | 5 | - |
| Stimulus delay | $t_{\text{sd}}$ | $\mathcal{U}(5, 25)$ | - | $\mathcal{U}(2, 8)$ | - |
| Decision duration | $t_{\text{dec}}$ | - | 20 | 10 | 1 |
| Input amplitude | $u_{\text{amp}}$ | 1 | 1 | $\mathcal{U}(0.5, 1.5)$ | 1 |
| Target amplitude | $\hat{z}_{\text{amp}}$ | 0.5 | 0.5 | 0.5 | $\{0.5, 2.0\}$ |

## 2 Task details

All task share a broad overall structure: a trial of length $T$ contains an initial "fixation" period without input of length $t_{\text{fix}}$, followed by the first input. During each input phase of duration $t_{\text{stim}}$, all or some of the inputs $u_i$ have a nonzero value with amplitude $u_{\text{amp}}$. Finally, there are distinct decision periods during which the target $\hat{z}$ is nonzero, with amplitude $\hat{z}_{\text{amp}}$. The decision periods are preceded by a decision delay, in which the loss is inactive, and which allows the output to converge to the target value. For the flip-flop task and the simple task, the loss is inactive outside of the decision periods; for the Mante and Romo tasks, all output channels are supposed to stay at zero until the beginning of the decision delay (the corresponding target is $\hat{z}_i = 0$ for all channels $i$). Below, we describe further details for each task. The parameters and their numerical values used in the simulations reported in the main text are summarized in Table S1.

**Flip-flop task**  During each trial, the network receives a number of short pulses of duration $t_{\text{stim}}$. During such a pulse, one input channel is set to $u_i(t) = s\, u_{\text{amp}}$, the others remain zero. The channel and sign $s \in \{\pm 1\}$ are chosen at random. After each pulse and a following delay period $t_{\text{delay}}$, a decision period starts (the loss is activated). During the decision period, the target value is set to $\hat{z}_i(t) = s\, \hat{z}_{\text{amp}}$. The other channel is supposed to remain silent, $\hat{z}_j(t) = 0$ for $j \neq i$. The decision period ends with the next pulse. The delays between stimuli $t_{\text{sd}}$ are drawn randomly. Note that the plotted trial time in Fig. 1 in the main text is $T = 100$, while training was done for $T = 50$.

**Mante task**  Each trial for the Mante task contains only a single, longer input period of duration $t_{\text{stim}}$. Half of the input channels correspond to the signal $u_i(t)$, the other half to a context variable $u_{N_s+i}(t)$, with number of signals $N_s = N_{\text{in}}/2$. The signals each consist of a constant mean and random noise part: $u_i(t) = u_{\text{amp}}[s_i + a_{\text{noise}}\eta_i(t)]$ with random sign $s_i \in \{\pm 1\}$ and Gaussian white noise $\eta_i(t)$. For our simulations, we chose the relative noise amplitude $a_{\text{noise}} = 0.05$. For the discretization, the white noise at time step $k$ is $\eta_{i,k} = n_{i,k}/\sqrt{\eta}$ with standard normal variable $n_{i,k} \sim \mathcal{N}(0, 1)$. During each trial, only a single context is active, $u_{N_s+i} = u_{\text{amp}}\delta_{i,j}$, where $j$ is chosen randomly from the number of inputs $N_s$. Outside of the input period, all mean values of $u_i$ are set to zero (the noise terms remain active). The input period is followed by a decision phase of length $t_{\text{dec}}$, with a delay $t_{\text{delay}}$ in between. During the decision period, the output is supposed to communicate the sign $s_j$ of the relevant input $j$. The target is constant: $\hat{z}(t) = \hat{z}_{\text{amp}}s_j$, and $\hat{z}_i(t) = 0$ for all $i \neq j$.

**Romo task**  For the Romo task, the RNN model has only one input channel, and each trial contains two input pulses of length $t_{\text{stim}}$ each. During the input pulses, the input is $u(t) = u_{\text{amp},1}$ and $u(t) = u_{\text{amp},2}$, with amplitudes drawn from a uniform distribution. Both input amplitudes are redrawn if their difference $|u_{\text{amp},1} - u_{\text{amp},2}|$ is below a minimal difference $u_{\text{min diff}} = 0.2$. The two pulses are separated by a random delay $t_{\text{sd}}$. The end of the second pulse is followed by a delay $t_{\text{delay}}$ and a decision period of length $t_{\text{dec}}$.

During the decision period, the output should indicate which input pulse was larger: $\hat{z}_j(t) = \hat{z}_{\mathrm{amp}}$ for $j = \arg\max_i(u_{\mathrm{amp,i}})$. The other output should remain at zero.

**Simple task**   The simple task only has a single input and output channel. The input is constant starting from the end of the fixation period: $u(t) = u_{\mathrm{amp}}$ for $t > t_{\mathrm{fix}}$. The decision period is a short interval at the end of the trial, $[T - t_{\mathrm{dec}}, T]$. The target value during the decision period is $\hat{z}(t) = \hat{z}_{\mathrm{amp}}$. There is no decision delay, and the input remains constant during the decision period. Hence, this task does not contain a memory element like the other three tasks.

# 3  Supplementary figures

Fig. S1: Scaling of learning dynamics with network size $N$ for all three nonlinear tasks and three different values of initial connectivity $g$ (indicated by line colors). Lines indicate average over 5 independent simulations, shades the standard deviation. Note the log-scale for networks size (x-axes). **(a-c)** Number of epochs until loss reached 5% of its initial value. **(d-f)** Frobenius norm of final connectivity changes $\Delta W$. **(g-i)** Frobenius norm of total connectivity $W = W_0 + \Delta W$. **(j-l)** Functional rank as defined in the main text (the rank at which truncation loss falls below 5% of the initial loss).

Fig. S2: Singular values (SVs) and eigenvalues (EVs) of RNNs trained for all three tasks with different initial connectivity strength $g \in \{0.0, 0.9, 1.8\}$. **(a-c)** First 40 SVs of the weight changes $\Delta W$ (top) and the final weight matrix $W = W_0 + \Delta W$ (bottom). Note the different y-scales: For $g = 0$ (darkest lines), the SVs in both plots are the same. For larger $g$, the SVs of $\Delta W$ tend to become smaller, while those of $W$ increase. **(d-l)** Eigenvalue spectra for $\Delta W$ (left) and $W$ (right). The x- and y-coordinates are the real and imaginary part, respectively. For $g = 0$, **(d-f)**, the EVs of $\Delta W$ and $W$ are the same. For $g > 0$, we plot the circles with radius $g$ for comparison. Inside this radius, the eigenvalues of $W_0$ are distributed uniformly [1]. Note that most EVs of $W$ still remain with in this circle. Parameters as in Fig. 1 of the main text, specifically $N = 256$.

Fig. S3: Evolution of SVs on log scale for the simple task, as a supplement to Fig. 3 of the main text. There, the SVs are shown on a linear scale, which does not allow to observe the evolution of any but the largest SVs. Our theory predicts only the first two SVs (dashed lines); any higher SVs are zero at order $\mathcal{O}(\tau^3)$. **(a,b)** Loss curves as a reference for the learning process. **(c,d)** First five SVs for $g = 0$. Note that the curves of $s_3$, $s_4$, and $s_5$ overlap. **(e,f)** First five SVs for $g = 0.6$.

Fig. S4: Exploding gradient when the real part of the largest eigenvalue $\lambda_1$ of $W$ crosses 1. For infinitely small learning rate $\eta$, the readout $z$ crosses the target value $\hat{z}$ before $\lambda_1$ crosses 1, so that learning stops. However, for a finite learning time, $z$ may become larger than an $\hat{z}$, and the divergent gradient may induce oscillations and failure of learning. This failure happens for large target values $\hat{z}$ and initial connectivity strength $g$, which promote the growth of $\lambda_1$. **(a,b)** Loss curves for two different target values and initial connectivity strengths. For $\hat{z} = 4$ and $g = 0.8$, the gradient diverges and learning stops. **(c-f)** Real parts of first five EVs $\lambda_i$ (order by decreasing real parts). Symbols at the end of each trajectory indicate the eigenvalues. In case of complex conjugates, the two corresponding $\lambda_i$ are written next to each other. The dashed grey line indicates the critical value $\Re\lambda = 1$ for which the gradients diverge. Parameters: $N = 256$, $\eta = \eta_0(1 - g^2)^2$ with $\eta_0 = 0.002, 0.001$ for $\hat{z} = 2, 4$, respectively. Task parameters as in the main text but with longer trial time, $T = 201$ (so that the network still converges to the fixed point despite the slower time scales).

Fig. S5: Example of learning only in presence of initial connectivity. For linear RNNs without initial connectivity, gradient descent-induced connectivity changes are always constructed from the input- and output vectors. If the space of these vectors is too small, learning fails. Here, we take the simple example of a linear network learning a cosine oscillation, starting from a fixed initial condition [see **(a)**]. The initial condition is set by a delta pulse through the input vector; otherwise, the input is zero. We set both input and output vector to **w**, so that there is only a single vector available. However, creating the necessary complex conjugate eigenvalues needs a rank-two connectivity and hence at least two different directions. Random initial connectivity enlarges the pool of available directions. **(a)** Output of networks at the end of training for three different values of $g$. Dashed line shows target $\hat{z}(t) = \cos(2\pi f t)$ with frequency $f = 0.2$. Learning failed for $g = 0$. For the other two values, the network finds a slightly unstable solution (perfect marginal stability is not achieved because of the limited trial time $T = 20$). **(b)** Loss over training epochs. **(c)** Imaginary part of largest eigenvalue $\lambda^{(1)}$, sorted by *imaginary parts*. **(d)** Real part of $\lambda^{(1)}$. The dashed lines show the real part of the largest eigenvalue sorted by real parts. For $g = 0$, no nonzero eigenvalue emerges throughout training. Parameters: $N = 256$, $\eta = (0.2, 0.15, 0.05)$ for $g = (0.0, 0.4, 0.8)$, respectively (adapted heuristically for smooth convergence); training for 1000 epochs (batch size = 1, since there is not stochastic part). Simulation step size was reduced to $\Delta t = 0.1$.

# 4 Expansion of linear learning

For the simple learning problem, the readout in the limit $t \to \infty$ is given by

$$z = \mathbf{w}^T (I - W)^{-1} \mathbf{m}\,. \tag{5}$$

The loss is quadratic: $L = (\hat{z} - z)^2 / 2$. The weights change according to the gradient of the loss w.r.t. to recurrent weights $W$, namely

$$\frac{\mathrm{d}W(\tau)}{\mathrm{d}\tau} = -\frac{\mathrm{d}L}{\mathrm{d}W} = [\hat{z} - z(\tau)] \left[I - W^T(\tau)\right]^{-1} \mathbf{w}\mathbf{m}^T \left[I - W^T(\tau)\right]^{-1}\,. \tag{6}$$

We expand these dynamics in orders of $\tau$. In the main text, we introduced the expansion

$$W(\tau) = \sum_{k=0}^{\infty} W_k \frac{\tau^k}{k!}\,, \tag{7}$$

with coefficients $W_k$ obtained from $\mathrm{d}^k W / \mathrm{d}\tau^k$ at $\tau = 0$.

## 4.1 First order

Because of the independence of $W_0$, $\mathbf{w}$, and $\mathbf{m}$, the initial readout $z_0$ is zero, and we directly obtain

$$W_1 = \hat{z} B^T \mathbf{w}\mathbf{m}^T B^T\,, \tag{8}$$

with $B = (I - W_0)^{-1}$. The weight changes linear in $\tau$ are

$$\Delta W(\tau) = \mathbf{u}_1 \mathbf{v}_1^T + \mathcal{O}(\tau^2)\,, \tag{9}$$

with

$$\mathbf{u}_1 = a_1 B^T \mathbf{w}\,, \qquad \mathbf{v}_1^T = a_1 \mathbf{m}^T B^T\,, \tag{10}$$

and the coefficient

$$a_1^2 = \tau \hat{z}\,. \tag{11}$$

Note that we chose to split the norm of the rank-one matrix equally between the two vectors, which simplifies notation later on. To compute the readout, we note that $W_1$ is a rank-one matrix. This allows us to apply the matrix inversion lemma (a.k.a. Sherman-Morrison formula; [2]): The matrix $I - W_0$ is invertible for $g < 1$, and subtracting a rank-one matrix $\mathbf{u}\mathbf{v}^T$ changes its inverse to

$$\left(I - W_0 - \mathbf{u}\mathbf{v}^T\right)^{-1} = B + \frac{1}{1 - \mathbf{v}^T B \mathbf{u}} B \mathbf{u}\mathbf{v}^T B\,, \tag{12}$$

To compute the readout at linear order, we first realize that the scalar product in the denominator in Eq. (12) vanishes:

$$\mathbf{v}_1^T B \mathbf{u}_1 = a_1^2 \mathbf{m}^T B^T B B^T \mathbf{w} = 0\,. \tag{13}$$

To show this, we note that $\mathbf{m}$ and $\mathbf{w}$ are independent of $M = B^T B B^T$, and therefore

$$\mathbb{E}\left[\mathbf{m}^T M \mathbf{w}\right] = \sum_{i=1}^{N} \sum_{j=1}^{N} \underbrace{\mathbb{E}[m_i w_j]}_{=0} \mathbb{E}[M_{ij}]\,. \tag{14}$$

The variance of $\mathbf{m}^T M \mathbf{w}$ is of order $1/N$, so that in the limit of $N \to \infty$, the term self-averages to zero. With this, we can compute the readout:

$$
\begin{aligned}
z &= \mathbf{w}^T \left( I - W_0 - \mathbf{u}_1 \mathbf{v}_1^T \right)^{-1} \mathbf{m} \\
&= \underbrace{\mathbf{w}^T B \mathbf{m}}_{=0} + \mathbf{w}^T B \mathbf{u}_1 \, \mathbf{v}_1^T B \mathbf{m} \\
&= \tau \hat{z} \, \mathbf{w}^T B B^T \mathbf{w} \, \mathbf{m}^T B^T B \mathbf{m} \\
&= \tau \hat{z} \beta^2 + \mathcal{O}(\tau^2) \,.
\end{aligned}
\tag{15}
$$

The term $\mathbf{w}^T B B^T \mathbf{w}$ (and likewise $\mathbf{m}^T B^T B \mathbf{m}$) has expectation

$$
\mathbb{E}\left[ \mathbf{w}^T B B^T \mathbf{w} \right] = \sum_{i=1}^{N} \sum_{j=1}^{N} \underbrace{\mathbb{E}[w_i w_j]}_{=\delta_{ij}/N} \mathbb{E}[(BB^T)_{ij}] = \frac{1}{N} \mathbb{E}[\mathrm{Tr}(BB^T)] = \beta \,.
\tag{16}
$$

The expected trace $\beta = 1/(1 - g^2)$ is computed in Section 5. Due to self-averaging in the limit $N \to \infty$, we omit the expectation.

The singular values of $W_1$ are the square roots of the eigenvalues of

$$
W_1 W_1^T = \hat{z}^2 B^T \mathbf{w} \mathbf{m}^T B^T B \mathbf{m} \mathbf{w}^T B \,.
\tag{17}
$$

Since this is again a rank-one matrix, we compute the only nonzero eigenvalue via the trace:

$$
s^2 = \mathrm{Tr}(W_1 W_1^T) = \hat{z}^2 \mathbf{w}^T B B^T \mathbf{w} \mathbf{m}^T B^T B \mathbf{m} = \hat{z}^2 \beta^2 \,.
\tag{18}
$$

The singular value, which is also the norm of $W_1$, is therefore

$$
s = ||W_1|| = \hat{z}\beta \,.
\tag{19}
$$

The learning time $\tau_1^*$ is the solution to the equation $z(\tau_1^*) = \hat{z}$, namely $\tau_1^* = 1/\beta^2$. The connectivity changes at this learning time are $\Delta W = \tau_1^* W_1$, with norm $||\Delta W|| = \tau_1^* ||W_1|| = \hat{z}/\beta$.

## 4.2 Second order

We again make use of the matrix inversion lemma, Eq. (12), and compute

$$
\begin{aligned}
W_2 &= \left. \frac{\mathrm{d}^2 W}{\mathrm{d}\tau^2} \right|_{\tau=0} \\
&= \left. \frac{\mathrm{d}}{\mathrm{d}\tau} \left[ (\hat{z} - z) \left( I - W_0 - \mathbf{u}_1 \mathbf{v}_1^T \right)^{-T} \mathbf{w} \mathbf{m}^T \left( I - W_0 - \mathbf{u}_1 \mathbf{v}_1^T \right)^{-T} \right] \right|_{\tau=0} \\
&= \left. \frac{\mathrm{d}}{\mathrm{d}\tau} \left[ (\hat{z} - z) B^T \left( I + \mathbf{v}_1 \mathbf{u}_1^T B^T \right) \mathbf{w} \mathbf{m}^T \left( I + B^T \mathbf{v}_1 \mathbf{u}_1^T \right) B^T \right] \right|_{\tau=0} \\
&= \left. \frac{\mathrm{d}}{\mathrm{d}\tau} \left[ (\hat{z} - \tau \hat{z} \beta^2) B^T \left( \mathbf{w} + \tau \hat{z} \beta B \mathbf{m} \right) \left( \mathbf{m}^T + \tau \hat{z} \beta \mathbf{w}^T B \right) B^T \right] \right|_{\tau=0} \\
&= \hat{z} \beta \, B^T \left[ -\beta \mathbf{w} \mathbf{m}^T + \hat{z} \left( \mathbf{w} \mathbf{w}^T B + B \mathbf{m} \mathbf{m}^T \right) \right] B^T \,.
\end{aligned}
\tag{20}
$$

We notice that the weight changes up to order $\mathcal{O}(\tau^2)$ can be written as the outer product of two vectors and is thus a rank-one matrix:

$$
\begin{aligned}
\Delta W &= \tau W_1 + \frac{\tau^2}{2} W_2 + \mathcal{O}(\tau^3) \\
&= B^T \left[ \left( \tau \hat{z} - \frac{\tau^2}{2} \hat{z}\beta^2 \right) \mathbf{w}\mathbf{m}^T + \frac{\tau^2}{2} \hat{z}^2 \beta \left( \mathbf{w}\mathbf{w}^T B + B\mathbf{m}\mathbf{m}^T \right) \right] B^T + \mathcal{O}(\tau^3) \\
&= B^T (a_2 \mathbf{w} + b_2 B\mathbf{m}) \left( a_2 \mathbf{m}^T + b_2 \mathbf{w}^T B^T \right) B^T + \mathcal{O}(\tau^3) \\
&= \mathbf{u}_2 \mathbf{v}_2^T + \mathcal{O}(\tau^3) \,,
\end{aligned}
\tag{21}
$$

with

$$
\mathbf{u}_2 = B^T (a_2 \mathbf{w} + b_2 B\mathbf{m}) \,, \qquad \mathbf{v}_2^T = \left( a_2 \mathbf{m}^T + b_2 \mathbf{w}^T B^T \right) B^T \,.
\tag{22}
$$

The coefficients are implicitly defined by

$$
a_2^2 = \tau \hat{z} - \frac{\tau^2}{2} \hat{z}\beta^2 \,, \qquad a_2 b_2 = \frac{\tau^2}{2} \hat{z}^2 \beta \,.
\tag{23}
$$

Note that the correction $b_2^2$ from completing the square is of order $\mathcal{O}(\tau^3)$.

Similarly to the first order, we can compute the readout $z$:

$$
z_2 = \frac{\mathbf{w}^T B \mathbf{u}_2 \mathbf{v}_2^T B \mathbf{m}}{1 - \mathbf{v}_2^T B \mathbf{u}_2} = a_2^2 \beta^2 + \mathcal{O}(\tau^3) \,,
\tag{24}
$$

with

$$
\mathbf{w}^T B \mathbf{u}_2 = \mathbf{v}_2^T B \mathbf{m} = a_2 \beta \,.
\tag{25}
$$

The denominator is of order $\mathcal{O}(\tau^2)$ and hence does not contribute to $z_2$:

$$
\begin{aligned}
\mathbf{v}_2^T B \mathbf{u}_2 &= \left( a_2 \mathbf{m}^T + b_2 \mathbf{w}^T B^T \right) B^T B B^T (a_2 \mathbf{w} + b_2 B\mathbf{m}) \\
&= 2 a_2 b_2 \gamma + \mathcal{O}(\tau^3) \,.
\end{aligned}
\tag{26}
$$

The random matrix term $\gamma = \mathbf{w}^T B B^T B B^T \mathbf{w} = \beta^4$ is compute Section 5. Terms of the form $\mathbf{m}^T M \mathbf{w}$, with $M$ constructed from $B$ and $B^T$ are zero due to the independence of all three quantities.

## 4.3 Third order

Since $\Delta W$ at order $\mathrm{O}(\tau^2)$ is a rank-1 matrix, we can use the same formalism as for the second order, cf. Eq. (20). We now only keep terms with $\tau^2$:

$$
\begin{aligned}
W_3 &= \left. \frac{\mathrm{d}^3 W}{\mathrm{d}\tau^3} \right|_{\tau=0} \\
&= \left. \frac{\mathrm{d}^2}{\mathrm{d}\tau^2} \left[ (\hat{z} - z) \left( I - W_0 - \mathbf{u}_2 \mathbf{v}_2^T \right)^{-T} \mathbf{w}\mathbf{m}^T \left( I - W_0 - \mathbf{u}_2 \mathbf{v}_2^T \right)^{-T} \right] \right|_{\tau=0} \\
&= \left. \frac{\mathrm{d}^2}{\mathrm{d}\tau^2} \left[ (\hat{z} - z) B^T \left( I + \mathbf{v}_2 \mathbf{u}_2^T B^T \right) \mathbf{w}\mathbf{m}^T \left( I + B^T \mathbf{v}_2 \mathbf{u}_2^T \right) B^T \right] \right|_{\tau=0} \\
&= \left. \frac{\mathrm{d}^2}{\mathrm{d}\tau^2} \left[ (\hat{z} - a_2^2 \beta^2) B^T \left[ \mathbf{w} + \beta B \left( a_2^2 \mathbf{m} + a_2 b_2 B^T \mathbf{w} \right) \right] \left[ \mathbf{m}^T + \beta \left( a_2^2 \mathbf{w}^T + a_2 b_2 \mathbf{m}^T B^T \right) B \right] B^T \right] \right|_{\tau=0} \\
&= \hat{z}\beta^2 B^T \left[ \beta^2 \mathbf{w}\mathbf{m}^T - 3\hat{z}\beta \left( \mathbf{w}\mathbf{w}^T B + B\mathbf{m}\mathbf{m}^T \right) + 2\hat{z}^2 B\mathbf{m}\mathbf{w}^T B + \hat{z}^2 \left( \mathbf{w}\mathbf{m}^T B^T B + B B^T \mathbf{w}\mathbf{m}^T \right) \right] B^T \,.
\end{aligned}
\tag{27}
$$

The changes up to order $\mathcal{O}(\tau^2)$ are now of rank two:

$$
\begin{aligned}
\Delta W &= \tau W_1 + \frac{\tau^2}{2} W_2 + \frac{\tau^3}{6} W_2 + \mathcal{O}(\tau^4) \\
&= B^T \Bigg[ \left( \tau \hat{z} - \frac{\tau^2}{2}\hat{z}\beta^2 + \frac{\tau^3}{6}\hat{z}\beta^4 \right) \mathbf{w}\mathbf{m}^T + \left( \frac{\tau^2}{2}\hat{z}^2\beta - \frac{\tau^3}{2}\hat{z}^2\beta^3 \right) \left( \mathbf{w}\mathbf{w}^T B + B\mathbf{m}\mathbf{m}^T \right) \\
&\quad + \frac{\tau^3}{3}\hat{z}^3\beta^2 B\mathbf{m}\mathbf{w}^T B + \frac{\tau^3}{6}\hat{z}^3\beta^2 \left( \mathbf{w}\mathbf{m}^T B^T B + BB^T \mathbf{w}\mathbf{m}^T \right) \Bigg] B^T + \mathcal{O}(\tau^4) \\
&= B^T \left( a_3\mathbf{w} + b_3 B\mathbf{m} + c_3 BB^T\mathbf{w} \right) \left( a_3\mathbf{m}^T + b_3\mathbf{w}^T B + c_3\mathbf{m}^T B^T B \right) B^T + \hat{b}_3^2 B^T B\mathbf{m}\mathbf{w}^T BB^T + \mathcal{O}(\tau^4) \\
&= \mathbf{u}_3\mathbf{v}_3^T + \hat{\mathbf{u}}_3\hat{\mathbf{v}}_3^T + \mathcal{O}(\tau^4) \,,
\end{aligned}
$$

$$(28)$$

with

$$
\mathbf{u}_3 = B^T \left( a_3\mathbf{w} + b_3 B\mathbf{m} + c_3 BB^T\mathbf{w} \right) \,, \tag{29}
$$

$$
\mathbf{v}_3^T = \left( a_3\mathbf{m}^T + b_3\mathbf{w}^T B + c_3\mathbf{m}^T B^T B \right) B^T \,, \tag{30}
$$

$$
\hat{\mathbf{u}}_3 = \hat{b}_3 B^T B\mathbf{m} \,, \tag{31}
$$

$$
\hat{\mathbf{v}}_3^T = \hat{b}_3 \mathbf{w}^T BB^T \,. \tag{32}
$$

The coefficients are implicitly defined by

$$
a_3^2 = \tau \hat{z} - \frac{\tau^2}{2}\hat{z}\beta^2 + \frac{\tau^3}{6}\hat{z}\beta^4 \,, \tag{33}
$$

$$
a_3 b_3 = \frac{\tau^2}{2}\hat{z}^2\beta - \frac{\tau^3}{2}\hat{z}^2\beta^3 \,, \tag{34}
$$

$$
a_3 c_3 = \frac{\tau^3}{6}\hat{z}^3\beta^2 \,, \tag{35}
$$

$$
b_3^2 = \frac{(a_3 b_3)^2}{a_3^2} = \frac{\tau^3}{4}\hat{z}^3\beta^2 \,, \tag{36}
$$

$$
\hat{b}_3^2 = \frac{\tau^3}{3}\hat{z}^3\beta^2 - b_3^2 = \frac{\tau^3}{12}\hat{z}^3\beta^2 \,. \tag{37}
$$

The remaining corrections $b_3 c_3$ and $c_3^2$ are of order $\mathcal{O}(\tau^4)$ or higher.

The changes $\Delta W$ can be written in a compact rank-two form:

$$
\Delta W(\tau) = \begin{bmatrix} \mathbf{u}_3 & \hat{\mathbf{u}}_3 \end{bmatrix} \begin{bmatrix} \mathbf{v}_3^T \\ \hat{\mathbf{v}}_3^T \end{bmatrix} + \mathcal{O}(\tau^4) = UV^T + \mathcal{O}(\tau^4) \,. \tag{38}
$$

With this, we compute the readout, using the matrix inversion lemma [2]:

$$
\begin{aligned}
z &= \mathbf{w}^T \left( I - W_0 - UV^T \right)^{-1} \mathbf{m} + \mathcal{O}(\tau^4) \\
&= \mathbf{w}^T \left[ B + BU \left( I_2 - V^T BU \right)^{-1} V^T B \right] \mathbf{m} + \mathcal{O}(\tau^4) \\
&= \mathbf{w}^T BU \left( I_2 - V^T BU \right)^{-1} V^T B\mathbf{m} + \mathcal{O}(\tau^4) \,.
\end{aligned}
$$

$$(39)$$

Here, $I_2$ is the $2 \times 2$ identity matrix. We compute the entries of $V^T B U$ up to $\mathcal{O}(\tau^3)$:

$$\mathbf{v}_3^T B \mathbf{u}_3 = 2 a_3 b_3 \gamma \,, \tag{40}$$

$$\mathbf{v}_3^T B \hat{\mathbf{u}}_3 = a_3 \hat{b}_3 \gamma \,, \tag{41}$$

$$\hat{\mathbf{v}}_3^T B \mathbf{u}_3 = a_3 \hat{b}_3 \gamma \,, \tag{42}$$

$$\hat{\mathbf{v}}_3^T B \hat{\mathbf{u}}_3 = 0 \,. \tag{43}$$

The factor $\gamma = \beta^4$ is computed in Section 5. Therefore,

$$I_2 - V^T B U = \begin{bmatrix} 1 - x & -y \\ -x & 1 \end{bmatrix} \,, \tag{44}$$

with $x = \mathbf{v}_3^T B \mathbf{u}_3$, and $y = \mathbf{v}_3^T B \hat{\mathbf{u}}_3$. Since $p$ and $q$ are $\mathcal{O}(\tau^2)$, we have

$$\left( I_2 - V^T B U \right)^{-1} = \frac{1}{1 - x - y^2} \begin{bmatrix} 1 & y \\ y & 1 - x \end{bmatrix} = \begin{bmatrix} 1 + x & y \\ y & 1 \end{bmatrix} + \mathcal{O}(\tau^4) \,. \tag{45}$$

To complete the evaluation of $z$, Eq. (39), we further compute $\mathbf{w}^T B U$ and $V^T B \mathbf{m}$:

$$\mathbf{w}^T B \mathbf{u}_3 = \mathbf{v}_3^T B \mathbf{m} = a_3 \beta + c_3 \gamma \,, \tag{46}$$

$$\mathbf{w}^T B \hat{\mathbf{u}}_3 = \hat{\mathbf{v}}_3^T B \mathbf{m} = 0 \,. \tag{47}$$

Hence,

$$\begin{aligned} z &= \begin{bmatrix} \mathbf{w}^T B \mathbf{u}_3 & \mathbf{w}^T B \hat{\mathbf{u}}_3 \end{bmatrix} \begin{bmatrix} 1 + x & y \\ y & 1 \end{bmatrix} \begin{bmatrix} \mathbf{v}_3^T B \mathbf{m} \\ \hat{\mathbf{v}}_3^T B \mathbf{m} \end{bmatrix} + \mathcal{O}(\tau^4) \\ &= (1 + x) \mathbf{w}^T B \mathbf{u}_3 \, \mathbf{v}_3^T B \mathbf{m} + \mathcal{O}(\tau^4) \\ &= (1 + 2 a_3 b_3 \gamma) \, (a_3 \beta + c_3 \gamma)^2 + \mathcal{O}(\tau^4) \\ &= \Big( 1 + \underbrace{2 a_3 b_3 \gamma}_{\mathcal{O}(\tau^2)} \Big) \Big( \underbrace{a_3^2 \beta^2}_{\mathcal{O}(\tau)} + \underbrace{2 a_3 c_3 \beta \gamma}_{\mathcal{O}(\tau^3)} + \underbrace{c_3^2 \gamma^2}_{\mathcal{O}(\tau^4)} \Big) + \mathcal{O}(\tau^4) \\ &= a_3^2 \beta^2 + 2 a_3 c_3 \beta \gamma + 2 a_3^2 a_3 b_3 \beta^2 \gamma + \mathcal{O}(\tau^4) \\ &= \hat{z} \left[ \beta^2 \tau - \frac{(\beta^2 \tau)^2}{2} + (1 + 8 \hat{z}^2 \beta) \frac{(\beta^2 \tau)^3}{6} \right] + \mathcal{O}(\tau^4) \,. \end{aligned} \tag{48}$$

The last lines are based on the implicit definitions of the coefficients $a_3$, $b_3$, and $c_3$ in Eqs. (33) to (35) and $\gamma = \beta^4$.

We end this section with looking at the special case $g = 0$. With $B = I$ and $\beta = 1$, the weight changes Eq. (28) simplify to

$$\begin{aligned} \Delta W &= \left( \tau \hat{z} - \frac{\tau^2}{2} \hat{z} + \frac{\tau^3}{2} \hat{z} \right) \mathbf{w} \mathbf{m}^T + \left( \frac{\tau^2}{2} \hat{z}^2 - \frac{\tau^3}{2} \hat{z}^2 \right) \left( \mathbf{w} \mathbf{w}^T + \mathbf{m} \mathbf{m}^T \right) + \frac{\tau^3}{3} \hat{z}^3 \mathbf{m} \mathbf{w}^T + \mathcal{O}(\tau^4) \\ &= \begin{bmatrix} \mathbf{w} & \mathbf{m} \end{bmatrix} \begin{bmatrix} A_{11} & A_{12} \\ A_{21} & A_{22} \end{bmatrix} \begin{bmatrix} \mathbf{w}^T \\ \mathbf{m}^T \end{bmatrix} \,, \end{aligned} \tag{49}$$

with

$$A_{11} = \frac{\hat{z}^2}{2} \left( \tau^2 - \tau^3 \right) + \mathcal{O}(\tau^4) \,, \tag{50}$$

$$A_{12} = \hat{z} \left( \tau - \frac{\tau^2}{2} + \frac{\tau^3}{6} (1 + 2 \hat{z}^2) \right) + \mathcal{O}(\tau^4) \,, \tag{51}$$

$$A_{21} = \frac{\hat{z}^3 \tau^3}{3} + \mathcal{O}(\tau^4) \,, \tag{52}$$

and $A_{22} = A_{11}$. Note that for $g = 0$, one can write the entire gradient descent dynamics in terms of the matrix $2 \times 2$ matrix $A$:

$$\frac{\mathrm{d}A}{\mathrm{d}\tau} = (\hat{z} - z) \left[ I + C^T \right] \begin{bmatrix} 1 \\ 0 \end{bmatrix} \begin{bmatrix} 0 & 1 \end{bmatrix} \left[ I + C^T \right] , \tag{53}$$

with

$$z = \begin{bmatrix} 1 & 0 \end{bmatrix} \left[ I + C \right] \begin{bmatrix} 0 \\ 1 \end{bmatrix} = C_{12} , \tag{54}$$

and

$$C = A(I - A)^{-1} . \tag{55}$$

With the symmetry $A_{11} = A_{22}$, this equation still has three degrees of freedom, and we were not able to find a closed form solution.

## 4.4 Singular values of weight changes

The singular values of $\Delta W$ are determined by the eigenvalues of $\Delta W^T \Delta W$ up to order $\mathcal{O}(\tau^3)$. For the rank-two matrix $\Delta W = UV^T$, these are the eigenvalues of the matrix

$$P = V^T V U^T U = \begin{bmatrix} p & q \\ q & r \end{bmatrix}^2 = \begin{bmatrix} p^2 + q^2 & q(p + r) \\ q(p + r) & q^2 + r^2 \end{bmatrix}^2 . \tag{56}$$

As before, we compute the coefficients up to order $\mathcal{O}(\tau^3)$:

$$p = \mathbf{u}_3^T \mathbf{u}_3 = \mathbf{v}_3^T \mathbf{v}_3 = a^2 \beta + (b^2 + 2ac)\gamma , \tag{57}$$

$$q = \mathbf{u}_3^T \hat{\mathbf{u}}_3 = \mathbf{v}_3^T \hat{\mathbf{v}}_3 = b\hat{b}\gamma , \tag{58}$$

$$r = \hat{\mathbf{u}}_3^T \hat{\mathbf{u}}_3 = \hat{\mathbf{v}}_3^T \hat{\mathbf{v}}_3 = \hat{b}^2 \gamma . \tag{59}$$

The squared singular values are therefore

$$s_\pm^2 = \frac{1}{2} \left( \mathrm{Tr}P \pm \sqrt{(\mathrm{Tr}P)^2 - 4|P|} \right) . \tag{60}$$

The terms are of order $p = \mathcal{O}(\tau)$ and $q, r = \mathcal{O}(\tau^3)$, so that

$$\mathrm{Tr} = p^2 + 2q^2 + r^2 = \mathcal{O}(\tau^2) , \tag{61}$$

$$|P| = (pr - q^2)^2 = \mathcal{O}(\tau^8) . \tag{62}$$

This means that the solutions have different orders:

$$s_+^2 = \mathrm{Tr}P - \frac{|P|}{\mathrm{Tr}P} , \tag{63}$$

$$s_-^2 = \frac{|P|}{\mathrm{Tr}P} . \tag{64}$$

Taking the square roots and sorting out the orders yields a linear first singular value,

$$s_+ = \frac{\hat{z}}{\beta} \left[ \beta^2 \tau - \frac{(\beta^2 \tau)^2}{2} + \left( 1 + \frac{7}{2} \hat{z}^2 \beta \right) \frac{(\beta^2 \tau)^3}{6} \right] . \tag{65}$$

The second singular value is cubic in learning time:

$$s_- = \hat{b}_{(3)}^2 \gamma = \hat{z}^3 \frac{(\beta^2 \tau)^3}{12} . \tag{66}$$

## 4.5 Effect of shuffling

Shuffling $W_0$ at the end of training destroys any correlation between $W_0$ and $W_1$, while keeping the same statistics. We denote that shuffled $W_0$ by $W_0^s$, and the corresponding inverse by $B^s = (1 - W_0^s)^{-1}$.

At first order, the shuffled readout is

$$
\begin{aligned}
z^s(\tau) &= \mathbf{w}^T (I - W_0^s - \tau_1^* W_1)^{-1} \mathbf{m} \\
&= \mathbf{w}^T \Big[ B^s + \frac{1}{1 - \underbrace{\mathbf{v}_1^T B^s \mathbf{u}_1}_{=0}} B^s \mathbf{u}_1 \mathbf{v}_1^T B^s \Big] \mathbf{m} \\
&= \underbrace{\mathbf{w}^T B^s \mathbf{m}}_{=0} + \mathbf{w}^T B^s \mathbf{u}_1 \, \mathbf{v}_1^T B^s \mathbf{m} \\
&= \tau \hat{z} \, \mathbf{w}^T B^s B^T \mathbf{w} \, \mathbf{m}^T B^T B^s \mathbf{m} \\
&= \tau \hat{z} + \mathcal{O}(\tau^2) \,.
\end{aligned}
\tag{67}
$$

The factor $\beta$ vanishes because

$$
\mathbb{E}\left[\mathbf{w}^T B^s B^T \mathbf{w}\right] = \sum_{i=1}^N \sum_{j=1}^N \sum_{k=1}^N \underbrace{\mathbb{E}[w_i w_k]}_{=\delta_{ik}/N} \mathbb{E}[B_{ij}^s] \mathbb{E}[B_{jk}^T] = \frac{1}{N} \sum_{i=1}^N \sum_{j=1}^N \underbrace{\mathbb{E}[B_{ij}^s]}_{=\delta_{ij}\left(1+\frac{1}{N}\right)} \underbrace{\mathbb{E}[B_{ji}^T]}_{=\delta_{ji}\left(1+\frac{1}{N}\right)} = 1 + \mathcal{O}(1/N) \,.
\tag{68}
$$

Inserting $\tau_1^* = 1/\beta^2$ into Eq. (67) yields $z^s(\tau_1^*) = \hat{z}/\beta^2$. The corresponding loss is

$$
L^s = \frac{1}{2}(\hat{z} - z^s(\tau_1^*))^2 = \frac{1}{2}\hat{z}^2 \left(1 - \frac{1}{\beta^2}\right)^2 = L_0 g^4 (2 - g^2)^2 \,,
\tag{69}
$$

with initial loss $L_0 = \hat{z}^2/2$.

For the third order, not all amplification is lost: Replacing $B$ with $B^s$ in the evaluation of $z$, Eq. (39) yields

$$
\begin{aligned}
z^s &= \mathbf{w}^T \left(I - W_0^s - UV^T\right)^{-1} \mathbf{m} + \mathcal{O}(\tau^4) \\
&= \mathbf{w}^T B^s U \left(I_2 - V^T B^s U\right)^{-1} V^T B^s \mathbf{m} + \mathcal{O}(\tau^4) \,.
\end{aligned}
\tag{70}
$$

We compute

$$
x^s = \mathbf{v}_3^T B^s \mathbf{u}_3 = a_3 b_3 \left(\mathbf{m}^T B^T B^s B^T B \mathbf{m} + \mathbf{w}^T B B^T B^s B^T \mathbf{w}\right) = 2 a_3 b_3 \beta^2 \,.
\tag{71}
$$

This is based on

$$
\begin{aligned}
\mathbb{E}\left[\mathbf{w}^T B B^T B^s B^T \mathbf{w}\right] &= \sum_{i=1}^N \sum_{j=1}^N \underbrace{\mathbb{E}[w_i w_j]}_{=\delta_{ij}/N} \mathbb{E}[(B B^T B^s B^T)_{ij}] \\
&= \frac{1}{N} \sum_{i,j,k,l} \mathbb{E}[B_{ij} B_{jk}^T B_{li}^T] \underbrace{\mathbb{E}[B_{kl}^s]}_{=\delta_{kl}\left(1+\frac{1}{N}\right)} \\
&= \frac{1}{N} \sum_{i,j,k,l} \mathbb{E}[B_{ij} B_{jk}^T B_{ki}^T] \underbrace{\mathbb{E}[B_{kl}^s]}_{=\delta_{kl}\left(1+\frac{1}{N}\right)} \\
&= \frac{1}{N} \mathbb{E}[\operatorname{Tr}(B B^T B^T)] = \beta^2 \,.
\end{aligned}
\tag{72}
$$

Similarly,

$$
y^s = \mathbf{v}_3^T B^s \hat{\mathbf{u}}_3 = \hat{\mathbf{v}}_3^T B^s \mathbf{u}_3 = a_3 \hat{b}_3 \beta^2 \,, \qquad \hat{\mathbf{v}}_3^T B^s \hat{\mathbf{u}}_3 = 0 \,,
\tag{73}
$$

and

$$\mathbf{w}^T B^s \mathbf{u}_3 = \mathbf{v}_3^T B^s \mathbf{m} = a_3 + c_3 \beta^2 \,, \tag{74}$$

$$\mathbf{w}^T B^s \hat{\mathbf{u}}_3 = \hat{\mathbf{v}}_3^T B^s \mathbf{m} = 0 \,. \tag{75}$$

The remaining parts of the calculation of $z$ are similar to the case without shuffling, and the corresponding result to Eq. (48) is:

$$
\begin{aligned}
z^s &= \begin{bmatrix} \mathbf{w}^T B^s \mathbf{u}_3 & \mathbf{w}^T B^s \hat{\mathbf{u}}_3 \end{bmatrix} \begin{bmatrix} 1 + x^s & y^s \\ y^s & 1 \end{bmatrix} \begin{bmatrix} \mathbf{v}_3^T B^s \mathbf{m} \\ \hat{\mathbf{v}}_3^T B^s \mathbf{m} \end{bmatrix} + \mathcal{O}(\tau^4) \\
&= (1 + x^s) \mathbf{w}^T B^s \mathbf{u}_3 \, \mathbf{v}_3^T B^s \mathbf{m} + \mathcal{O}(\tau^4) \\
&= (1 + 2a_3 b_3 \beta^2) \, (a_3 + c_3 \beta^2)^2 + \mathcal{O}(\tau^4) \\
&= \Big( 1 + \underbrace{2a_3 b_3 \beta^2}_{\mathcal{O}(\tau^2)} \Big) \Big( \underbrace{a_3^2}_{\mathcal{O}(\tau)} + \underbrace{2a_3 c_3 \beta^2}_{\mathcal{O}(\tau^3)} + \underbrace{c_3^2 \beta^4}_{\mathcal{O}(\tau^4)} \Big) + \mathcal{O}(\tau^4) \\
&= a_3^2 + 2a_3 c_3 \beta^2 + 2a_3^2 a_3 b_3 \beta^2 + \mathcal{O}(\tau^4) \\
&= \frac{\hat{z}}{\beta^2} \left[ \beta^2 \tau - \frac{(\beta^2 \tau)^2}{2} + \left( 1 + 2\hat{z}^2 \left( 1 + \frac{3}{\beta} \right) \right) \frac{(\beta^2 \tau)^3}{6} \right] + \mathcal{O}(\tau^4) \,.
\end{aligned}
\tag{76}
$$

A comparison with Eq. (48) shows that the first and second order terms are decreased by $1/\beta^2$. However, the third order term has a correction to this, similar to the learning time $\tau^*$.

## 5 Traces

Here we compute traces appearing in our learning problem:

$$\frac{1}{N} \mathrm{Tr}(B) = 1 \,, \tag{77}$$

$$\frac{1}{N} \mathrm{Tr}(BB^T) = \beta \,, \tag{78}$$

$$\frac{1}{N} \mathrm{Tr}(BBB^T) = \beta^2 \,, \tag{79}$$

$$\frac{1}{N} \mathrm{Tr}(BB^T BB^T) = \gamma = \beta^4 \,, \tag{80}$$

with $B = (I - J)^{-1}$ and $\beta = \frac{1}{1 - g^2}$. The matrix $J$ is a Gaussian random matrix whose entries are drawn independently from $\mathcal{N}(0, g^2/N)$. We denote $W_0 = J$ in order to avoid the extra index.

The traces generally stem from scalar products of the form $\mathbf{a}^T M \mathbf{a}$, where the entries of the random vector $\mathbf{a}$ are drawn from $\mathcal{N}(0, 1/N)$, and the matrix $M$ is independent of $\mathbf{a}$. In particular, any combinations of the matrices $B$ are independent of $\mathbf{a}$, since they only contain the random matrix $J$. Because of this independence, we have

$$\mathbb{E}\left[\mathbf{a}^T M \mathbf{a}\right] = \sum_{i,j=1}^{N} \mathbb{E}[a_i M_{ij} a_j] = \sum_{i,j=1}^{N} \underbrace{\mathbb{E}[a_i a_j]}_{=\delta_{ij}/N} \mathbb{E}[M_{ij}] = \mathbb{E}\left[\frac{\mathrm{Tr} M}{N}\right] \,. \tag{81}$$

Computing the traces above and showing the self-averaging quality of the terms is a matter of counting the number of contributing combinations of $J$ and $J^T$. Our results are based on expanding $B$ into a geometric series

$$B = I + \sum_{K=1}^{\infty} J^K \,. \tag{82}$$

## 5.1 $\mathrm{Tr}(B)$

We start with the trace of $B$ alone:

$$\mathbb{E}\left[\frac{\mathrm{Tr}B}{N}\right] = 1 + \sum_{K=1}^{\infty} \frac{1}{N}\sum_{i=1}^{N}\mathbb{E}\left[(J^K)_{ii}\right] = 1 + \mathcal{O}\left(\frac{1}{N}\right). \tag{83}$$

We show why the sum vanishes with $N$. For $K=1$, the entries $J_{ii}$ have expectation 0. For $K=2$, the independence of elements of $J$ yields

$$\frac{1}{N}\sum_{i=1}^{N}\mathbb{E}\left[(J^2)_{ii}\right] = \frac{1}{N}\sum_{i,j=1}^{N}\mathbb{E}\left[J_{ij}J_{ji}\right] = \frac{1}{N}\sum_{i\neq j}\underbrace{\mathbb{E}\left[J_{ij}\right]\mathbb{E}\left[J_{ji}\right]}_{=0} + \frac{1}{N}\sum_{i}\underbrace{\mathbb{E}\left[J_{ii}^2\right]}_{=g^2/N} = \mathcal{O}\left(\frac{1}{N}\right). \tag{84}$$

The second term vanishes because there are only $N$ terms, but the factor $1/N$ before the sum and the contribution $g^2/N$ together yield $1/N^2$. This observation can be generalized to higher $K$:

$$\frac{1}{N}\sum_{i=1}^{N}\mathbb{E}\left[(J^K)_{ii}\right] = \frac{1}{N}\sum_{i_1,i_2,\ldots,i_K}\mathbb{E}\left[J_{i_1 i_2}J_{i_2 i_3}\ldots J_{i_K i_1}\right] = \frac{1}{N}\sum_{i}\underbrace{\mathbb{E}\left[J_{ii}^{K/2}\right]}_{=\mathcal{O}\left(N^{K/2}\right)} = \mathcal{O}\left(\frac{1}{N^{K/2-1}}\right). \tag{85}$$

There are $K$ different indices. Because each index appears once as a first and once as a second index, the attempt to form pairs directly results in setting all indices equal.

## 5.2 $\mathrm{Tr}(BB^T)$

The situation changes when introducing $B^T$. We can write

$$BB^T = \sum_{K,L=0}^{\infty} J^K J^{TL}, \tag{86}$$

where the transpose $T$ and power $L$ commute. We compute the trace again term by term, starting at $K=L=1$:

$$\frac{1}{N}\sum_{i=1}^{N}\mathbb{E}\left[(JJ^T)_{ii}\right] = \frac{1}{N}\sum_{i,j}\mathbb{E}\left[J_{ij}J_{ji}^T\right] = \frac{1}{N}\sum_{i,j}\underbrace{\mathbb{E}\left[J_{ij}^2\right]}_{=g^2/N} = g^2. \tag{87}$$

For general $K, L \geq 1$, we have

$$\begin{aligned}
\frac{1}{N}\sum_{i=1}^{N}\mathbb{E}\left[(J^K J^{TL})_{ii}\right] &= \frac{1}{N}\sum_{i_1,\ldots i_K}\sum_{j_1,\ldots j_L}\mathbb{E}\left[J_{i_1 i_2}J_{i_2 i_3}\ldots J_{i_K j_1}J_{j_1 j_2}^T J_{j_2 j_3}^T \ldots J_{j_L i_1}^T\right] \\
&= \frac{1}{N}\sum_{i_1,\ldots i_K}\sum_{j_1,\ldots j_L}\mathbb{E}\left[J_{i_1 i_2}J_{i_2 i_3}\ldots J_{i_K j_1}J_{j_2 j_1}J_{j_3 j_2}\ldots J_{j_1 i_L}^T\right].
\end{aligned} \tag{88}$$

We need to form pairs of indices. To simplify the discussion, we write the sequence of index pairs alone:

$$\begin{bmatrix}i_1\\i_2\end{bmatrix}\begin{bmatrix}i_2\\i_3\end{bmatrix}\ldots\begin{bmatrix}i_{K-1}\\i_K\end{bmatrix}\begin{bmatrix}i_K\\i_1\end{bmatrix}\begin{bmatrix}j_2\\j_1\end{bmatrix}\begin{bmatrix}j_3\\j_2\end{bmatrix}\ldots\begin{bmatrix}i_1\\j_L\end{bmatrix}. \tag{89}$$

There are $K + L$ indices, and we need to form $(K + L)/2$ distinct pairs of index pairs. Each index constraint reduces the entire term by a factor of $1/N$. Because of the additional factor $1/N$ in front of the sum, we can have only $(K + L)/2 - 1$ index constraints. The question becomes one of counting the number of possible combinations.

The expression above indicates that the only relevant term needs to constrain $i_K = j_2$. Under this

condition, we have

$$\begin{bmatrix} i_1 \\ i_2 \end{bmatrix} \begin{bmatrix} i_2 \\ i_3 \end{bmatrix} \cdots \begin{bmatrix} i_{K-1} \\ i_K \end{bmatrix} \begin{bmatrix} i_K \\ j_1 \end{bmatrix} \begin{bmatrix} i_K \\ j_1 \end{bmatrix} \begin{bmatrix} j_3 \\ i_K \end{bmatrix} \cdots \begin{bmatrix} i_1 \\ j_L \end{bmatrix} . \tag{90}$$

The two middle terms drop and the new middle pairs show the same configuration. One can proceed iteratively with this scheme until reaching the right or left end (depending on $min(K, L)$). In fact, if $L > K$, then

$$\underbrace{\begin{bmatrix} i_1 \\ i_2 \end{bmatrix} \begin{bmatrix} i_2 \\ i_3 \end{bmatrix} \cdots \begin{bmatrix} i_{K-1} \\ i_K \end{bmatrix} \begin{bmatrix} i_K \\ j_1 \end{bmatrix} \begin{bmatrix} i_K \\ j_1 \end{bmatrix} \begin{bmatrix} i_{K-1} \\ i_K \end{bmatrix} \cdots \begin{bmatrix} i_1 \\ i_2 \end{bmatrix}}_{\text{paired with } K-1 \text{ constraints}} \begin{bmatrix} j_{K+1} \\ i_1 \end{bmatrix} \begin{bmatrix} j_{K+2} \\ j_{K+1} \end{bmatrix} \cdots \begin{bmatrix} i_1 \\ j_L \end{bmatrix} . \tag{91}$$

The non-paired terms need $L - K$ additional constraints, so that the entire term only gives a contribution of $\mathcal{O}(1/N^{(L-K)/2-1})$. This and a similar argument for $K > L$ shows that we need $K = L$. In that case, there are $K - 1 = (K + L)/2 - 1$ constraints and the term contributes at order $\mathcal{O}(1)$. We summarize with

$$\mathbb{E}\left[\frac{\text{Tr}(J^K J^{TL})}{N}\right] = g^{2K}\delta_{KL} + \mathcal{O}\left(\frac{1}{N}\right) . \tag{92}$$

For the entire matrix $BB^T$, this leads to

$$\mathbb{E}\left[\frac{\text{Tr}(BB^T)}{N}\right] = \sum_{K,L=1}^{\infty} \mathbb{E}\left[\frac{\text{Tr}(J^K J^{TL})}{N}\right] = \sum_{K=1}^{\infty} g^{2K} + \mathcal{O}\left(\frac{1}{N}\right) = \frac{1}{1-g^2} + \mathcal{O}\left(\frac{1}{N}\right) . \tag{93}$$

Note that the correction terms remain finite under the infinite sums for $K$ and $L$ because they scale with $g^{K+L}$ and we chose $g < 1$.

## 5.3 $\text{Tr}(BBB^T)$

For $\text{Tr}(BBB^T)$, the arguments go in parallel to the previous discussion. Indeed, we have

$$\begin{aligned}
\mathbb{E}\left[\frac{\text{Tr}(BBB^T)}{N}\right] &= \sum_{K,L,M=1}^{\infty} \mathbb{E}\left[\frac{\text{Tr}(J^K J^L J^{TM})}{N}\right] \\
&= \sum_{K,L,M=1}^{\infty} g^{2M}\delta_{K+L,M} \\
&= \sum_{M=0}^{\infty} g^{2M} \underbrace{\sum_{K=0}^{\infty}\sum_{L=0}^{\infty} \delta_{K+L,M}}_{=\sum_{K=0}^{M} 1} \\
&= \sum_{M=0}^{\infty} g^{2M}(M+1) \\
&= \frac{1}{(1-g^2)^2} .
\end{aligned} \tag{94}$$

plus an order $\mathcal{O}(1/N)$ correction.

## 5.4 $\text{Tr}(BB^T BB^T)$

For $\mathbb{E}[\text{Tr}(BB^T BB^T)/N]$, we first compute trace of the components $J^i J^{Tj} J^k J^{Tl}$. Similar to the cases discussed before, we need to constrain indices to create equal index pairs. The index pairs before any constraints can be written as

$$\begin{bmatrix} i_1 & i_2 & \dots & i_i & j_2 & j_3 & \dots & k_1 & k_1 & k_2 & \dots & k_k & l_2 & l_3 & \dots & i_1 \\ i_2 & i_3 & \dots & j_1 & j_1 & j_2 & \dots & j_j & k_2 & k_3 & \dots & l_1 & l_1 & l_2 & \dots & l_l \end{bmatrix} . \tag{95}$$

There are $n = i + j + k + l$ summation indices, and each pair contributes with a factor $g^2/N$. Together with the additional factor $1/N$, we can thus have at most $n/2 - 1$ constraints. Note that like before, the number of transposed matrices must equal that of the non-transposed, $i + k = j + l$, so that $n$ is even. A smaller number of constraints is not sufficient, so that the question becomes: How many different sets of $n/2 - 1$ constraints lead to $n/2$ pairs of index pairs?

We start with $i = j = k = l = 1$. The corresponding index pairs are

$$\begin{bmatrix} i_1 & k_1 & k_1 & i_1 \\ j_1 & j_1 & l_1 & l_1 \end{bmatrix}. \tag{96}$$

One can see that there are two possible combinations to create two pairs: $i_1 = k_1$ and $j_1 = l_1$, which yield

$$\begin{bmatrix} i_1 & i_1 & i_1 & i_1 \\ j_1 & j_1 & l_1 & l_1 \end{bmatrix}, \qquad \begin{bmatrix} i_1 & k_1 & k_1 & i_1 \\ j_1 & j_1 & j_1 & j_1 \end{bmatrix}. \tag{97}$$

Therefore, there are 2 combinations. An index-counting argument like before generalizes this result, showing that the number of combinations is equal to

$$c_{ijkl} = 1 + \min(i, j, k, l). \tag{98}$$

We prove this statement by induction: Let $i = \min(i, j, k, l)$ without loss of generality (since the trace is cyclic). We rewrite the index pairs Eq. (95) and color cases were two upper or lower indices are equal without any constraints:

$$\begin{bmatrix} i_1 & i_2 & \dots & i_i & j_2 & j_3 & \dots & k_1 & k_1 & k_2 & \dots & k_k & l_2 & l_3 & \dots & i_1 \\ i_2 & i_3 & \dots & j_1 & j_1 & j_2 & \dots & j_j & k_2 & k_3 & \dots & l_1 & l_1 & l_2 & \dots & l_l \end{bmatrix}. \tag{99}$$

We next separate two cases: Case 1, $i_i = j_2$, and Case 2, $i_i \neq j_2$. In Case 1, the index pairs with the blue $j$s above become equal:

$$\begin{bmatrix} i_1 & i_2 & \dots & i_{i-1} & j_2 & j_2 & j_3 & \dots & k_1 & k_1 & k_2 & \dots & k_k & l_2 & l_3 & \dots & i_1 \\ i_2 & i_3 & \dots & j_2 & j_1 & j_1 & j_2 & \dots & j_j & k_2 & k_3 & \dots & l_1 & l_1 & l_2 & \dots & l_l \end{bmatrix}. \tag{100}$$

We can take these pairs out, and the remaining indices read

$$\begin{bmatrix} i_1 & i_2 & \dots & i_{i-1} & j_3 & j_4 & \dots & k_1 & k_1 & k_2 & \dots & k_k & l_2 & l_3 & \dots & i_1 \\ i_2 & i_3 & \dots & j_2 & j_2 & j_3 & \dots & j_j & k_2 & k_3 & \dots & l_1 & l_1 & l_2 & \dots & l_l \end{bmatrix}, \tag{101}$$

where we colored the $j_2$ blue again. We now have $(i', j', k', l') = (i - 1, j - 1, k, l)$ indices, with $\min(i', j', k', l') = i - 1$. According to our induction hypothesis, there are $c_{i'j'k'l'} = 1 + i - 1 = i$ different sets of $n/2 - 2$ constraints. Adding the constraint of Case 1, $i_i = j_2$ yields the expected number of $n/2 - 1$ constraints.

It remains to show that Case 2 allows for exactly one set of $n/2 - 1$ constraints. Because $i_i \neq j_2$ in Eq. (99), we need to have a pair at the red $i_1$; otherwise, one needs $n/2$ constraints. The pair at $i_1$ requires $l_l = i_2$, and dropping the newly formed pair yields

$$\begin{bmatrix} i_2 & i_3 & \dots & i_i & j_2 & j_3 & \dots & k_1 & k_1 & k_2 & \dots & k_k & l_2 & l_3 & \dots & i_2 \\ i_2 & i_3 & \dots & j_1 & j_1 & j_2 & \dots & j_j & k_2 & k_3 & \dots & l_1 & l_1 & l_2 & \dots & l_{l-1} \end{bmatrix}. \tag{102}$$

We follow the same argumentation, constraining $l_{l-1} = i_3, \dots, l_{2+l-i} = i_i$. We arrive at

$$\begin{bmatrix} i_i & j_2 & j_3 & \dots & k_1 & k_1 & k_2 & \dots & k_k & l_2 & \dots & i_i \\ j_1 & j_1 & j_2 & \dots & j_j & k_2 & k_3 & \dots & l_1 & l_1 & \dots & l_{1+l-i} \end{bmatrix}. \tag{103}$$

Further setting $l_{1+l-i} = j_1$ and dropping the induced pair leads to

$$\begin{bmatrix} j_2 & j_3 & \dots & k_1 & k_1 & k_2 & \dots & k_k & l_2 & \dots & j_1 \\ j_1 & j_2 & \dots & j_j & k_2 & k_3 & \dots & l_1 & l_1 & \dots & l_{l-i} \end{bmatrix}. \tag{104}$$

This is equal to the case $J^{Tj}J^kJ^{T(l-i)}$. By the cyclic nature of the trace, this is equal to the case $J^{T(j+l-i)}J^k$. As discussed above, Section 5.2, only one set of $(j+l-i+k)/2-1$ constraints leads to a full separation into pairs. Note that if $i = l$, the last set of indices, Eq. (104), looks slightly different, but yields the same result.

Counting the number of constraints in Case 2 yields $1+i-2+1+(j+l-i+k)/2-1 = n/2-1$. Since there is no other combination for Case 2, the total number of constraint combinations is precisely $i+1 = 1 + \min(i,j,k,l) = c_{ijkl}$.

We return to the trace, which contains the factors $g^2$:

$$\mathbb{E}\left[\frac{\mathrm{Tr}(J^iJ^{Tj}J^kJ^{Tl})}{N}\right] = g^{2(i+k)}\,\delta_{i+k,j+l}\,c_{ijkl}\,. \tag{105}$$

We now evaluate the sums over $i,j,k,l$, starting with fixed $i$:

$$\sum_{j,k,l=1}^{\infty}\mathbb{E}\left[\frac{\mathrm{Tr}(J^iJ^{Tj}J^kJ^{Tl})}{N}\right] = \sum_{j,k,l=1}^{\infty} g^{2(i+k)}\,\delta_{i+k,j+l}\,c_{ijkl}\,. \tag{106}$$

We split the summation into different regimes:

$$\sum_{j,k,l=1}^{\infty} g^{2(i+k)}\,\delta_{i+k,j+l}\,c_{ijkl} = \sum_{\substack{j,l \\ j+l\geq i}}\sum_{k=1}^{\infty} g^{2(i+k)}\,\delta_{k,j+l-i}\,c_{ijkl}$$

$$= \sum_{\substack{j,l \\ j\geq i \\ l\geq i}} g^{2(j+l)}\,(i+1) + \sum_{\substack{j,l \\ j+l\geq i \\ \min(j,l)<i}} g^{2(j+l)}\,c_{ij(j+l-i)l} \tag{107}$$

$$= a + b + c + d\,,$$

where we split the second summand of the second-last line into two parts. The parts are:

$$a = \sum_{\substack{j,l \\ j\geq i \\ l\geq i}} g^{2(j+l)}\,(i+1) = (i+1)\sum_{j=i}^{\infty}(i+1)\left(\sum_{j=i}^{\infty}g^{2j}\right)^2 = \frac{(i+1)g^{4i}}{(1-g^2)^2}\,, \tag{108}$$

$$b = \sum_{\substack{j,l \\ j<i \\ l<i \\ j+l\geq i}} g^{2(j+l)}\,(j+l-i+1) = \frac{g^{2i}}{(1-g^2)^3}\left[i(1+g^{2i})(1-g^2)-(1-g^{2i})(1+g^2)\right]\,, \tag{109}$$

$$c = \sum_{\substack{j,l \\ j\geq i \\ l<i}} g^{2(j+l)}\,(l+1) = \sum_{j=i}^{\infty}\sum_{l=0}^{i-1}g^{2l}\,(l+1) = \frac{g^{2i}}{(1-g^2)^3}\left[1-g^{2i}-g^{2i}i(1-g^2)\right]\,, \tag{110}$$

$$d = \sum_{\substack{j,l \\ j<i \\ l\geq i}} g^{2(j+l)}\,(j+1) = c\,. \tag{111}$$

Joining all terms yields

$$\sum_{j,k,l=1}^{\infty}\mathbb{E}\left[\frac{\mathrm{Tr}(J^iJ^{Tj}J^kJ^{Tl})}{N}\right] = \frac{(i+1)g^{2i}}{(1-g^2)^2}\,. \tag{112}$$

Finally, we sum over $i$:

$$\mathbb{E}\left[\frac{\mathrm{Tr}(BB^T BB^T)}{N}\right] = \sum_{i,j,k,l=1}^{\infty} \mathbb{E}\left[\frac{\mathrm{Tr}(J^i J^{Tj} J^k J^{Tl})}{N}\right] = \sum_{i=0}^{\infty} \frac{(i+1)g^{2i}}{(1-g^2)^2} = \frac{1}{(1-g^2)^4}\,. \tag{113}$$

We return to the trace, which is therefore

$$\mathbb{E}\left[\frac{\mathrm{Tr}(J^i J^{Tj} J^k J^{Tl})}{N}\right] = g^{2(i+k)}\,\delta_{i+k,j+l}\,c_{ijkl}\,. \tag{114}$$

We now evaluate the sums over $i,j,k,l$, starting with fixed $i$:

$$\sum_{j,k,l=1}^{\infty} \mathbb{E}\left[\frac{\mathrm{Tr}(J^i J^{Tj} J^k J^{Tl})}{N}\right] = \sum_{j,k,l=1}^{\infty} g^{2(i+k)}\,\delta_{i+k,j+l}\,c_{ijkl}\,. \tag{115}$$

We split the summation into different regimes:

$$\begin{aligned}
\sum_{j,k,l=1}^{\infty} g^{2(i+k)}\,\delta_{i+k,j+l}\,c_{ijkl} &= \sum_{\substack{j,l \\ j+l\ge i}} \sum_{k=1}^{\infty} g^{2(i+k)}\,\delta_{k,j+l-i}\,c_{ijkl} \\
&= \sum_{\substack{j,l \\ j\ge i \\ l\ge i}} g^{2(j+l)}\,(i+1) + \sum_{\substack{j,l \\ j+l\ge i \\ \min(j,l)<i}} g^{2(j+l)}\,c_{ij(j+l-i)l} \\
&= a + b + c + d\,,
\end{aligned} \tag{116}$$

where we split the second summand of the second-last line into two parts. The parts are:

$$a = \sum_{\substack{j,l \\ j\ge i \\ l\ge i}} g^{2(j+l)}\,(i+1) = (i+1)\sum_{j=i}^{\infty}(i+1)\left(\sum_{j=i}^{\infty} g^{2j}\right)^2 = \frac{(i+1)g^{4i}}{(1-g^2)^2}\,, \tag{117}$$

$$b = \sum_{\substack{j,l \\ j<i \\ l<i \\ j+l\ge i}} g^{2(j+l)}\,(j+l-i+1) = \frac{g^{2i}}{(1-g^2)^3}\left[i(1+g^{2i})(1-g^2) - (1-g^{2i})(1+g^2)\right]\,, \tag{118}$$

$$c = \sum_{\substack{j,l \\ j\ge i \\ l<i}} g^{2(j+l)}\,(l+1) = \sum_{j=i}^{\infty}\sum_{l=0}^{i-1} g^{2l}\,(l+1) = \frac{g^{2i}}{(1-g^2)^3}\left[1 - g^{2i} - g^{2i}i(1-g^2)\right]\,, \tag{119}$$

$$d = \sum_{\substack{j,l \\ j<i \\ l\ge i}} g^{2(j+l)}\,(j+1) = c\,. \tag{120}$$

Joining all terms yields

$$\sum_{j,k,l=1}^{\infty} \mathbb{E}\left[\frac{\mathrm{Tr}(J^i J^{Tj} J^k J^{Tl})}{N}\right] = \frac{(i+1)g^{2i}}{(1-g^2)^2}\,. \tag{121}$$

Finally, we sum over $i$:

$$\mathbb{E}\left[\frac{\mathrm{Tr}(BB^T BB^T)}{N}\right] = \sum_{i,j,k,l=1}^{\infty} \mathbb{E}\left[\frac{\mathrm{Tr}(J^i J^{Tj} J^k J^{Tl})}{N}\right] = \sum_{i=0}^{\infty} \frac{(i+1)g^{2i}}{(1-g^2)^2} = \frac{1}{(1-g^2)^4}\,, \tag{122}$$

Fig. S6: Details for 2-layer LSTM model trained on a sentiment analysis task. **(a, b)** Training and validation loss and accuracy over epochs. **(c-f)** Singular values (SVs) of the input and recurrent weights in both layers.

which is the statement we wanted to prove.

# 6 Details of sentiment analysis task

For the sentiment analysis task in the results section, we trained a 2-layer LSTM model on the Standford Sentiment Treebank with binary labels (SST-2) [7]. The dataset consists of sentences from movie reviews which are labeled positive or negative. Sentences have on average 20 words, and there are 6920 training and 872 validation examples. We tokenized the sentences with the scaCy tokenizer [3]. We further used a pretrained word embedding (GloVe, [6]) with dimension $N_{\text{in}} = 100$. The word embedding was kept fixed during training.

Each LSTM layer had $N = 256$ units. All weights and biases were initialized from the uniform distribution $\mathcal{U}(-a, a)$, where $a = \sqrt{1/N}$, except for input weights of layer 1, where $a = \sqrt{1/N_{\text{in}}}$. During training, all weights and biases were updated with Adam on a binary cross entropy loss, as implemented in PyTorch [5]. We set the learning rate to $0.01/N$, and all other parameters at their default values. We additionally applied dropout with probability 0.5 to all hidden states. We trained the model for 500 epochs, each epoch iterating over the entire data set with batches of 64 sentences.

To evaluate the performance after truncation, we separated the weights into recurrent and input weights. Because the LSTM for the four different gates are concatenated, the input weights of layer 1 have shape $4N \times N_{\text{in}}$, all other weights have shape $4N \times N$. We simultaneously truncated the recurrent weights of both layers and the input weights of layer 2, i.e., all blocks with shape $4N \times N$ This specific choice did not alter the qualitative result, namely that truncating the changes $\Delta W$ and $\Delta U$ at a given rank produces a much smaller decrease in performance than truncating the full weights $W = W_0 + \Delta W$ and $U = U_0 + \Delta U$.

Note that we chose the learning rate to be sufficiently small so that learning dynamics were smooth. With higher learning rates and rugged loss curves, we observed that changes $\Delta W$ would replace the initial connectivity, and the effective rank was much higher. Further note that other hyperparameters, such as L2 regularization on the weights, may also change the picture.