[Reviews · NeurIPS 2020]

Review 1

Summary and Contributions: The authors study how a given RNN architecture deviates from its initial random weights over training. Surprisingly, they find that in 3 very simple tasks, the deviation from the initial values is very low rank (rank 6 at most for an RNN with 256 units). Then the authors proceed to analytically study why this phenomena emerges, in a simplified setup (linear RNNs).

Strengths: - The authors make a surprising finding. When you decompose an RNNs weight matrix into its initial value and the deviation from it during training, the deviation (\delta W) can have very low rank. - Authors study the phenomena mathematically in a simplified setting. - The implications of such a phenomena (if it holds in realistic setups) are of VERY HIGH importance for the model compression community. I would be willing to considerably improve my rating if the authors can show that this phenomena happens in more realistic scenarios. - While the analytical study is nice, it takes too much space. I would focus more on experimentally analyzing it to argue it's a phenomena worth studying more in-depth.

Weaknesses: - There are no experiment in more complex problems, which reduces the impact of the work considerably.

Correctness: I haven't checked the math thoroughly, but it seems correct at first sight.

Clarity: The paper was mostly clear. Section 3 (the mathematical analysis) was harder to follow, but this is because of the complexity of it, not because of lack of clarity. Other than that, authors should mention the size of the RNN in the main text and not just the caption, as this is a key part to understand the experiments.

Relation to Prior Work: Authors have missed two very related works: 'Measuring the Intrinsic Dimension of Objective Landscapes': They show good performance can be achieved in complex tasks, when optimizing networks across low-dimensional subspaces. 'On the Effectiveness of Low-Rank Matrix Factorization for LSTM Model Compression': They perform SVD on the weight matrices of LSTMs for model compression. They show that in some cases, the dimensionality can be drastically decreased, without affecting the performance of the model too much.

Reproducibility: Yes

Additional Feedback: I would strongly recommend that authors try to apply their approach to LSTMs in standard RNN benchmarks, like language modelling. It has been shown that in this cases SVD factorizations can find decent low-rank representations. However, as authors point out in here, W = W_0 + \delta W, and given that W_0 is random, we can't expect W to be very low-rank. Thus, I'm optimistic \delta W will have a considerably lower rank than W, which would have serious implications for model compression (as W_0 can be generated from a pseudo-random number generator). Also, do you think that the optimizer may play a role in this? Given that Adam has momemtum, consecutive updates are highly correlated, which may lead to the low dimensionality of \delta W. Updated Review: The new results in SST-2 are impressive and very promising. I have updated my score accordingly.


Review 2

Summary and Contributions: POST REBUTTAL COMMENT ---- I thank the authors for the additional results in the rebuttal, and think that this paper is a good submission. I leave my score unchanged. ------ This paper examines the relationship between initial connectivity, task structure, and changes in connectivity in RNNs, by making the hypothesis low-dimensional tasks induce low-rank changes in connectivity, which in turn allows to explain the phenomenon of accelerated learning in the presence of random initial connectivity. One studies the influence of the connectivity strength of the initialization on the learning dynamics as well as the final connectivity matrix by proving a series of novel claims (analytically and experimentally) highlighting how the learning dynamics and the final connectivity are heavily influenced by the choice of initial connectivity. For the experimental part, the hypotheses are validated on three low-dimensional toy tasks inspired by the neuroscience literature. For the analytical part, the simplified analytically tractable case of a linear RNN on a simplified task is considered, where one quantifies the degree of accelerated learning as a function of the initial connectivity strength.

Strengths: The paper proves a diverse set of novel claims in regards to gradient-based learning dynamics in RNNs experimentally as well as analytically. It provides a useful framework that could be used to understand the interplay between task dimension, initial connectivity and rank of the weight matrix changes as well as their influence on the structure of the final weight matrix in broader settings of gradient-based learning in neural networks.

Weaknesses: One could have examined how exactly the claims are changing for more complex higher-dimensional tasks.

Correctness: The experimental methodology as well as the mathematical analysis both make sense and seem correct.

Clarity: The paper is very well written with a clear storyline. The experiments are clearly laid out and the analysis is rigorously and thoroughly explained.

Relation to Prior Work: The paper is put into context with respect to prior related work and seems sufficiently novel.

Reproducibility: Yes

Additional Feedback:


Review 3

Summary and Contributions: Recent studies on computation by recurrent neural networks assume that the computation performed by the system is due to a low-rank component in the dense connectivity matrix. In this work, the authors show for simple tasks that a gradient-decent algorithm, which is not constrained to operate in low-dimension, results in a low-rank change to the connectivity matrix. This low-rank solution is achieved with or without full-rank initial random connectivity. The authors further show that initial random connectivity can improve learning by shortening the learning time. By considering a linear network, the authors could derive an analytical expression for the loss and singular values of the learned weights across training. They show that the leaning follows nontrivial dynamics. Finally, they show that the simplified linear model is a good predictor for the behavior of more complex tasks and nonlinear networks.

Strengths: This work offers interesting new observations: * The authors show the benefits of training near the transition to chaos — something previously observed in RNN training but not rigorously shown through analytics. * They present evidence that the learned low-rank structure of the recurrent connectivity is correlated with the initial random connectivity. * An interesting finding is that even for a simple linear system with a fixed target, the training dynamics depend on the input's value in a nonlinear way. * The work provides excellent analytical treatment using a toy model that leads to interpretable results. The theory is verified with a close match to the toy model. Furthermore, a good match with more complex tasks indicates the right choice of a toy model.

Weaknesses: * The tasks studied in this work are all simple tasks that come down to converging to a fixed point. While this is the assumption of several recent studies in the field, it is a limited view of problems tackled by RNNs. I don't see any reason to assume that the low-rank connectivity assumption would extend to complex tasks where the target is not a simple fixpoint. For example, problems that require backpropagation through time to learn. I think the scope should be stated more clearly in the paper. * The analysis relies on power-law expansion in training time $\tau$. Many of the conclusions are made on the fixed point where $\beta^2\tau=1$. The authors note that the time always appears with $\beta^2\tau$, in Eq. (13)-(15). The authors use the expansion to conclude that the learning converges within $1/\beta^2$. It is not clear how the authors make this assumption about the convergence when it is clearly outside the scope of their approximation. Numerics indeed seems to validate the result, and I don't s reason to doubt the final result, but the validity of the derivations is not clear to me.

Correctness: The claims and methods are correct and the theory agrees well with numerical investigations.

Clarity: The paper is well written and should be accessible to a broad audience.

Relation to Prior Work: Yes.

Reproducibility: Yes

Additional Feedback: * [lines 263-264]: the authors claims that their expansion reveals a temporal hierarchy in the solution. A similar temporal hierarchy was observed and studied in a single layer perceptron. The emergence of temporal hierarchy was shown to be related to the structure of the input [Yoshida and Okada, NeurIPS 2019, Glodt et al. 2019 arxiv:1909.11500]. Could this be an interpretation of the temporal hierarchy here? ***** Pos author's response feedback ***** I am happy to see that the authors addressed the main concerns I had. I maintain the view that this work is interesting and novel, and I keep the original high score I gave.


Review 4

Summary and Contributions: This paper is a theoretical study of the dynamics of learning in RNNs. The study focuses on the relationship between the learnt recurrent connectivity of a network and the underlying task, and the way in which training shapes this relation. The authors first evaluate nonlinear RNNs trained on three example neuroscience-inspired tasks and make the following key observations: 1. Learnt connectivity changes are small (in Frobenius norm) compared to the initial connectivity 2. Training induced change in connectivity is of low-rank even though training is unconstrained 3. Amplitude and geometry of these low-rank changes depend on the initial connectivity 4. Strength of random initial connectivity (g) affects learning dynamics (convergence time first decreases with g and then increases again as networks transition to chaotic activity) To understand the mechanisms underlying these observations, the authors study a linear RNN performing a simple reaching task for a constant input signal. This simple setting allows them to perform a theoretical analysis of the network’s learning dynamics. They demonstrate that the connectivity changes are spanned by a small number of directions specified by the input and readout vectors, and random initial connectivity enlarges the pool of available directions. Further, they derive a Taylor series type expansion in learning time for the learnt weights. Using this expansion they show that some components in the connectivity only grow after others are present, which induces a temporal hierarchy in the learning dynamics.

Strengths: This work presents a purely theoretical study of learning dynamics in RNNs. The authors acknowledge that there are no new algorithms, tasks, or data sets introduced. This work however is a potentially important theoretical study. Although the main analytical results are obtained for a simple linear RNN, the approach stresses a dynamical perspective on learning which is important when trying to understand learning in (deep/) biological networks. [Update post-feedback. I appreciate that the authors tried their analysis on a new task. My score remains a solid accept.]

Weaknesses: I do have a concern about the expansion in eq(7), which requires learning time $$\tau$$ to be small. For what range of values of $$\tau$$ would this local approximation hold? (In figure 3b, learning converges for $$\tau$$ around 1)

Correctness: The derivations look right (although I only skimmed through the supplementary material) In Fig 3 and 4, the analytical results nicely match their empirical observations.

Clarity: Writing is very clear. Figures and figure labels are very clear as well. Minor: figure 1b, label should be \hat{z} instead of \hat{z}_1 ? might be useful to mention that (i) hats correspond to target variables, and (ii) eta is learning rate in the caption of fig 1.

Relation to Prior Work: Analytical approach is similar to the study of learning dynamics in feedforward nets: Saxe et al. A mathematical theory of semantic development in deep neural networks. 2019. This is cited in the paper.

Reproducibility: Yes

Additional Feedback:

[Author Response · NeurIPS 2020]

We thank the reviewers for their positive and constructive feedback on the paper. There are two main comments which
we will address below. We also addressed all the other comments, and they will appear in the revised version.

**Power law expansion** We thank the reviewers for highlighting
this point, as it led to a better formulation of the expansion. We
now expand in $\tilde{\tau} = \beta^2 \tau$ instead of in $\tau$. Figure R1 shows that for
low target values $\hat{z}$ the curves for different $g$-s collapse, indicating
that the relation between $g$ and learning time holds. Note that full
convergence occurs for $\tilde{\tau} > 1$, which is beyond the scope of the
expansion, but the majority of learning takes place before that time.

For large $\hat{z}$, $\tilde{\tau}$ remains small throughout training, indicating that the
expansion is valid. In this case, however, the curves only collapse
for the initial training phase. This deviation is due to higher-order
terms of the expansion, and is in the direction of decreasing training
time for increasing $g$ values – consistent with our main finding.
Specifically, expressing the third-order prediction for $z(\tau)$, Eq.
(15), in terms of $\tilde{\tau}$ shows this trend:

$$z(\tilde{\tau}) = \hat{z}\left[\tilde{\tau} - \frac{\tilde{\tau}^2}{2} + (1 + 8\hat{z}^2\beta)\frac{\tilde{\tau}^3}{6}\right] . \qquad (R1)$$

Fig. R1: Loss over effective learning time for the linear problem and different values of initial connectivity strength $g$ and target value $\hat{z}$ (cf. Fig. 3 of the main text). The curves collapse for $\hat{z} = 0.5$. For $\hat{z} = 2$, both the numerical results (full) and our third-order prediction (dashed)) separate at the end of training. The first-order prediction (dotted), $L(\tilde{\tau}) = L_0(1 - \tilde{\tau})^2$, has no dependence on $g$ other than through $\tilde{\tau}$.

**More complex tasks and network compression** We thank the reviewers for suggesting to study more complex tasks
and the topic of network compression. We trained an LSTM network on the NLP task of sentiment analysis (Fig. R2).
As in our paper, we found that the resulting changes to the network weights are of low rank. Furthermore, when we
truncate the *changes* in connectivity, a rank 10 matrix is sufficient to achieve full performance. This is compared
to a rank 200 matrix when trying to compress the full connectivity. We are not experts in NLP, and realize that this
preliminary result we obtained in a few days is not the end of the story. For the revised version, we will study the effect
of higher performing networks, non-binary NLP tasks and different word embeddings among others.

Note that one may not observe this behavior for any task and network off the shelf. In particular, the learning rate is
often chosen so high that learning dynamics become highly rugged. In such cases, we repeatedly observed weight
changes to be of much higher rank and effectively replacing any initial connectivity. Here we choose a sufficiently small
learning rate so that learning dynamics were smooth. Other hyperparameters, such as L2 regularization on the weights,
may also change the picture.

For more complex tasks, the emergence of low-rank structure cannot be explained by the small number of in- and
output vectors. Instead, we expect the (statistical) task structure to determine the rank, like recently established for
feed-forward networks (e.g., Advani & Saxe, 2017, Lampinen & Ganguli, 2019, and the works cited by Reviewer #3).

**Other points** Very briefly, our results do not depend on the Adam optimizer or model size. Our results rely on BPTT
(in particular for the Romo task), and hold for non fixed-point tasks (e.g., producing a periodic output).

Fig. R2: Low-rank changes for a 2-layer LSTM model trained on a sentiment analysis task, the Standford Sentiment Treebank with binary labels (SST-2). The pretrained word embedding (GloVe) with dimension $N_{in}$ was kept fixed during training, all other parameters were updated with Adam on a binary cross-entropy loss. (a) Training and validation accuracy over epochs. Final validation accuracy marked as a baseline for panel b. (b) Validation accuracy after truncating the lower singular values of connectivity. We either truncate $W$ directly, or apply truncation only to $\Delta W$ while keeping $W_0$. (c) Singular values (SVs) of the recurrent weights in the second layer. The initial, random $W_0$ is full rank, and the final $W$ visibly differs from it only for the first SVs. The changes, $\Delta W$, are approximately low-rank. Note that the LSTM weights for the four different gates are concatenated ($4N \times N$ matrices). SVs for the other (layer, input) weight matrices look similar. Parameters: Embedding and hidden dimension: $N_{in} = 100$, $N = 256$, learning rate $0.01/N$, dropout probability 0.5. Weight initialization: $\mathcal{U}(-a, a)$, where $a = \sqrt{1/N}$, except for input weights of layer 1, where $a = \sqrt{1/N_{in}}$.

[Meta-Review · NeurIPS 2020]

This is a nice illuminating study into the dynamics of how RNNs learn. The reviewers were all very positive about this paper, for the following reasons: 1) it’s very well-written and presented, 2) makes clear theoretical contributions and insights into the dynamics of RNNs throughout training, and 3) provides a nice analytical treatment and useful framework that could be applied in a more general sense. The rebuttal included new analyses on a more complex task, SST-2, with impressive results. I suggest the authors try to fit these into the main text, if possible. The findings should be of broad interest to the NeurIPS audience, and I strongly recommend accept.